# Homo-oligomerization of the human adenosine A$_{2A}$ receptor is driven by the intrinsically disordered C-terminus

**Khanh Dinh Quoc Nguyen[1], Michael Vigers[2], Eric Sefah[3], Susanna Seppälä[2], Jennifer Paige Hoover[1], Nicole Star Schonenbach[2], Blake Mertz[3], Michelle Ann O'Malley[2]\*, Songi Han[1,2]\***

[1]Department of Chemistry and Biochemistry, University of California, Santa Barbara, Santa Barbara, United States; [2]Department of Chemical Engineering, University of California, Santa Barbara, Santa Barbara, United States; [3]C. Eugene Bennett Department of Chemistry, West Virginia University, Morgantown, United States

**Abstract** G protein-coupled receptors (GPCRs) have long been shown to exist as oligomers with functional properties distinct from those of the monomeric counterparts, but the driving factors of oligomerization remain relatively unexplored. Herein, we focus on the human adenosine A$_{2A}$ receptor (A$_{2A}$R), a model GPCR that forms oligomers both in vitro and in vivo. Combining experimental and computational approaches, we discover that the intrinsically disordered C-terminus of A$_{2A}$R drives receptor homo-oligomerization. The formation of A$_{2A}$R oligomers declines progressively with the shortening of the C-terminus. Multiple interaction types are responsible for A$_{2A}$R oligomerization, including disulfide linkages, hydrogen bonds, electrostatic interactions, and hydrophobic interactions. These interactions are enhanced by depletion interactions, giving rise to a tunable network of bonds that allow A$_{2A}$R oligomers to adopt multiple interfaces. This study uncovers the disordered C-terminus as a prominent driving factor for the oligomerization of a GPCR, offering important insight into the effect of C-terminus modification on receptor oligomerization of A$_{2A}$R and other GPCRs reconstituted in vitro for biophysical studies.

**\*For correspondence:**
momalley@engineering.ucsb.edu (MAO'M);
songi@chem.ucsb.edu (SH)

**Competing interests:** The authors declare that no competing interests exist.

## Introduction

G protein-coupled receptors (GPCRs) have long been studied as monomeric units, but accumulating evidence demonstrates that these receptors can also form homo- and hetero-oligomers with far-reaching functional implications. The properties emerging from these oligomers can be distinct from those of the monomeric protomers in ligand binding (*El-Asmar et al., 2005*; *Casadó-Anguera et al., 2016*; *Guitart et al., 2014*; *Yoshioka et al., 2001*), G protein coupling (*Cristóvão-Ferreira et al., 2013*; *Cordomí et al., 2015*; *González-Maeso et al., 2007*; *Lee et al., 2004*; *Rashid et al., 2007*), downstream signaling (*Liu et al., 2016*; *Hilairet et al., 2003*; *Rozenfeld and Devi, 2007*; *Borroto-Escuela et al., 2010*), and receptor internalization/desensitization (*Ecke et al., 2008*; *Stanasila et al., 2003*; *Faklaris et al., 2015*). With the vast number of genes identified in the human genome (*Takeda et al., 2002*), GPCRs are able to form a daunting number of combinations with unprecedented functional consequences. The existence of this intricate network of interactions among GPCRs presents major challenges and opportunities for the development of novel therapeutic approaches (*Dorsam and Gutkind, 2007*; *Farran, 2017*; *Schonenbach et al., 2015*; *Ferré et al., 2014*; *Bräuner-Osborne et al., 2007*; *George et al., 2002*). Hence, it is crucial to identify the driving factors of GPCR oligomerization, such that this process can be more deliberately controlled to facilitate structure-function studies of GPCRs.

GPCR oligomers with multiple interfaces (*Song et al., 2020*; *Ghosh et al., 2014*; *Periole et al., 2012*; *Fanelli and Felline, 2011*; *Liu et al., 2012*) can give rise to myriad ways by which these complexes can be formed and their functions modulated. In the crystal structure of the turkey $\beta_1$-adrenergic receptor ($\beta_1$AR), the receptor appears to dimerize via two different interfaces, one formed via TM4/TM5 (transmembrane domains 4/5) and the other via TM1/TM2/H8 (helix 8) contacts (*Huang et al., 2013*). Similarly, in the crystal structure of the antagonist-bound $\mu$-opioid receptor ($\mu$-OR), the protomers also dimerize via two interfaces; however, only one of them is predicted to induce a steric hindrance that prevents activation of both protomers (*Manglik et al., 2012*), hinting at interface-specific functional consequences. A recent computational study predicted that the adenosine $A_{2A}$ receptor ($A_{2A}$R) forms homodimers via three different interfaces and that the resulting dimeric architectures can modulate receptor function in different or even opposite ways (*Fanelli and Felline, 2011*). All the above-mentioned interfaces are symmetric, meaning that the two protomers are in face-to-face orientations, hence forming strictly dimers. Asymmetric interfaces, reported in $M_3$ muscarinic receptor (*Thorsen et al., 2014*), rhodopsin (*Fotiadis et al., 2006*; *Fotiadis et al., 2003*; *Liang et al., 2003*), and opsin (*Liang et al., 2003*), are in contrast formed with the protomers positioning face-to-back, possibly enabling the association of higher-order oligomers.

Not only do GPCRs adopt multiple oligomeric interfaces, but various studies also suggest that these interfaces may dynamically rearrange to activate receptor function (*Xue et al., 2015*). According to a recent computational study, $A_{2A}$R oligomers can adopt eight different interfaces that interconvert when the receptor is activated or when there are changes in the local membrane environment (*Song et al., 2020*). Similarly, a recent study that combined experimental and computational data proposed that neurotensin receptor 1 (NTS$_1$R) dimer is formed by 'rolling' interfaces that coexist and interconvert when the receptor is activated (*Dijkman et al., 2018*). Clearly, meaningful functional studies of GPCRs require exploring their dynamic, heterogeneous oligomeric interfaces.

The variable nature of GPCR oligomeric interfaces suggests that protomers of GPCR oligomers may be connected by tunable interactions. In this study, we explore the role of an intrinsically disordered region (IDR) of a model GPCR that could engage in diverse non-covalent interactions, such as electrostatic interactions, hydrogen bonds, or hydrophobic interactions. These non-covalent interactions are readily tunable by external factors, such as pH, salts, and solutes, and further can be entropically enhanced by depletion interactions (*Asakura and Oosawa, 1958*; *Yodh et al., 2001*; *Marenduzzo et al., 2006*), leading to structure formation and assembly (*Milles et al., 2018*; *Wicky et al., 2017*; *Szasz et al., 2011*; *Goldenberg and Argyle, 2014*; *Qin and Zhou, 2013*; *Cino et al., 2012*; *Soranno et al., 2014*; *Zosel et al., 2020*). In a system where large protein molecules and small solute particles typically coexist in solution, assembly of the protein molecules causes their excluded volumes to overlap and the solvent volume accessible to the non-protein solutes to increase, raising the entropy of the system. The type and concentration of solutes or ions can also remove water from the hydration shell around the proteins, further enhancing entropy-driven protein-protein association in what is known as the hydrophobic effect (*Tanford, 1980*; *Tanford, 1978*; *Pratt and Chandler, 1977*; *van der Vegt et al., 2017*). This phenomenon is applied in the precipitation of proteins upon addition of so-called salting-out ions according to the Hofmeister series (*Hofmeister, 1888*; *Hyde et al., 2017*; *Yang, 2009*). The ability of IDRs to readily engage in these non-covalent interactions motivates our focus on the potential role of IDRs in driving GPCR oligomerization.

The cytosolic carboxy (C-)terminus of GPCRs is usually an IDR (*Tovo-Rodrigues et al., 2014*; *Jaakola et al., 2005*). Varying in length among different GPCRs, the C-terminus is commonly removed in structural studies of GPCRs to enhance receptor stability and conformational homogeneity. A striking example is $A_{2A}$R, a model GPCR with a particularly long, 122-residue, C-terminus that is truncated in all published structural biology studies (*Song et al., 2020*; *Fanelli and Felline, 2011*; *García-Nafría et al., 2018*; *Sun et al., 2017*; *Lebon et al., 2011*; *Xu et al., 2011*; *Doré et al., 2011*; *Jaakola et al., 2008*; *Carpenter et al., 2016*; *Hino et al., 2012*). However, evidence is accumulating that such truncations—shown to affect GPCR downstream signaling (*Koretz et al., 2021*; *Navarro et al., 2018a*; *Jain and McGraw, 2020*)—may abolish receptor oligomerization (*Schonenbach et al., 2016*; *Svetlana and Devi, 1997*). A study using immunofluorescence has demonstrated that C-terminally truncated $A_{2A}$R does not show protein aggregation or clustering on the cell surface, a process readily observed in the wild-type form (*Burgueño et al., 2003*). Our recent study employing a tandem three-step chromatography approach uncovered the

impact of a single-residue substitution of a C-terminal cysteine, C394S, in reducing the receptor homo-oligomerization in vitro (*Schonenbach et al., 2016*). In the context of heteromerization, mass spectrometry and pull-down experiments have demonstrated that $A_{2A}R$-$D_2R$ dimerization occurs via direct electrostatic interactions between the C-terminus of $A_{2A}R$ and the third intracellular loop of $D_2R$ (*Ciruela et al., 2004*). These results all suggest that the C-terminus may participate in $A_{2A}R$ oligomer formation. However, no studies to date have directly and systematically investigated the role of the C-terminus, or any IDRs, in GPCR oligomerization.

This study focuses on the homo-oligomerization of the human adenosine $A_{2A}R$, a model GPCR, and seeks to address (i) whether the C-terminus engages in $A_{2A}R$ oligomerization, and if so, (ii) whether the C-terminus forms multiple oligomeric interfaces. We use size-exclusion chromatography (SEC) to assess the oligomerization levels of $A_{2A}R$ variants with strategic C-terminal modifications: mutations of a cysteine residue C394 and a cluster of charged residues $^{355}ERR^{357}$, as well as systematic truncations at eight different sites along its length. We complemented our experimental study with an independent molecular dynamics (MD) simulation study of $A_{2A}R$ dimers of five C-terminally truncated $A_{2A}R$ variants designed to mirror the experimental constructs. We furthermore examined the oligomerization level of select C-terminally modified $A_{2A}R$ variants under conditions of varying ionic strength ranging from 0.15 to 0.95 M. To verify whether the $A_{2A}R$ oligomer populations are thermodynamic products, we performed a series of SEC analyses on SEC-separated monomer and dimer/oligomer populations to observe their repopulation into monomer and dimer/oligomer populations. Finally, to test whether the C-termini directly and independently promote $A_{2A}R$ oligomerization, we recombinantly expressed the entire $A_{2A}R$ C-terminal segment sans the transmembrane portion of the receptor and investigated its solubility and assembly properties with increasing ion concentration and temperature. This is the first study designed to uncover the role of the intrinsically disordered C-terminus on the oligomerization of a GPCR.

## Results

This study systematically investigates the role of the C-terminus on $A_{2A}R$ oligomerization and the nature of the involved interactions through strategic mutations and truncations at the C-terminus as well as modulation of the ionic strength of solvent. All experiments were done at 4°C unless stated otherwise. The experimental assessment of $A_{2A}R$ oligomerization relies on SEC analysis.

### SEC quantifies $A_{2A}R$ oligomerization

We performed SEC analysis on a mixture of ligand-active $A_{2A}R$ purified from a custom synthesized antagonist affinity column (*Figure 1—figure supplement 1A*). Distinct oligomeric species were separated and eluted in the following order: high-molecular-weight (HMW) oligomer, dimer, and monomer (*Figure 1* and *Figure 1—figure supplement 1B*). This peak assignment has been verified with SEC-MALS (multi-angle light scattering) experiments, as detailed in a previous publication (*Schonenbach et al., 2016*). The population of each oligomeric species was quantified as the integral of each Gaussian from a multiple-Gaussian curve fit of the SEC signal. The reported standard errors were calculated from the variance of the fit that do not correspond to experimental errors (see *Supplementary file 1* and *Figure 1—figure supplement 2* for SEC data corresponding to all $A_{2A}R$ variants in this study). As this study sought to identify the factors that promote $A_{2A}R$ oligomerization, the populations with oligomeric interfaces (i.e., dimer and HMW oligomer) were compared with those without such interfaces (i.e., monomer). Hence, the populations of the HMW oligomer and dimer were expressed relative to the monomer population in arbitrary units as monomer-equivalent concentration ratios, henceforth referred to as population levels (*Figure 1*).

### C-terminal amino acid residue C394 contributes to $A_{2A}R$ oligomerization

To investigate whether the C-terminus of $A_{2A}R$ is involved in receptor oligomerization, we first examined the role of residue C394 as a previous study demonstrated that the mutation C394S dramatically reduced $A_{2A}R$ oligomer levels (*Schonenbach et al., 2016*). The C394S mutation was replicated in our experiments, alongside other amino acid substitutions for the cysteine, namely alanine, leucine, methionine, or valine, generating five $A_{2A}R$-C394X variants. The HMW oligomer and dimer levels of $A_{2A}R$ wild-type (WT) were compared with those of the $A_{2A}R$-C394X variants. We found that

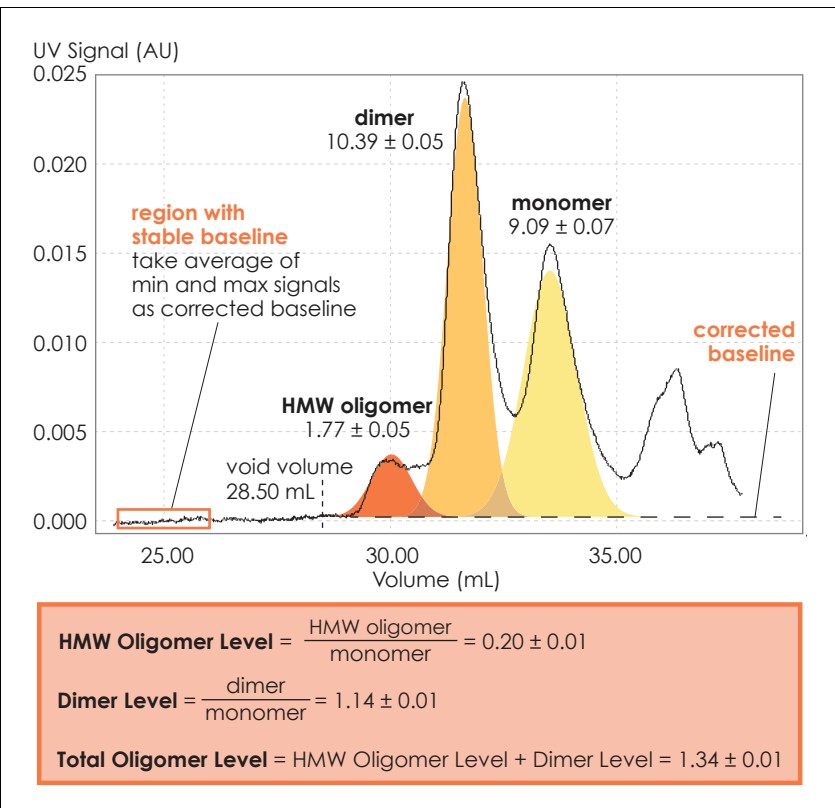

**Figure 1.** Method for collecting size-exclusion chromatography (SEC) data and assessing $A_{2A}R$ oligomerization. The SEC data is recorded every second as absorbance at 280 nm. The baseline is corrected to ensure uniform fitting and integration across the peaks. The areas under the curve, resulting from a multiple-Gaussian curve fit, express the population of each oligomeric species. The reported standard errors of integration are within a 95% confidence interval and are calculated from the variance of the fit, not experimental errors. The levels of high-molecular-weight oligomer and dimer are expressed relative to the monomeric population in arbitrary units. A representative calculation defining the oligomer levels is given in the box.

The online version of this article includes the following source data and figure supplement(s) for figure 1:

**Figure supplement 1.** The purity and identity of $A_{2A}R$ are confirmed with total protein stain and western blot.

**Figure supplement 1—source data 1.** Raw representative total protein stain of $A_{2A}R$-WT during purification.

**Figure supplement 1—source data 2.** Labeled representative total protein stain of $A_{2A}R$-WT during purification.

**Figure supplement 1—source data 3.** Raw representative western blot of $A_{2A}R$-WT during purification.

**Figure supplement 1—source data 4.** Labeled representative western blot of $A_{2A}R$-WT during purification.

**Figure supplement 1—source data 5.** Raw representative western blot of $A_{2A}R$-WT during size-exclusion chromatography separation.

**Figure supplement 1—source data 6.** Labeled representative western blot of $A_{2A}R$-WT during size-exclusion chromatography separation.

**Figure supplement 2.** Size-exclusion chromatographic traces and data distribution of all A2AR variants used in the main text of this study.

**Figure supplement 2—source data 1.** Raw size-exclusion chromatography data of five experimental replicates of $A_{2A}R$-WT.

the dimer level of $A_{2A}R$-WT was significantly higher than that of the $A_{2A}R$-C394X variants (WT: 1.14; C394X: 0.24–0.57; *Figure 2A*). A similar result, though less pronounced, was observed when the HMW oligomer and dimer levels were considered together (WT: 1.34; C394X: 0.59–1.21; *Figure 2A*). This suggests that residue C394 plays a role in $A_{2A}R$ oligomerization, and even more prominently in $A_{2A}R$ dimerization.

To test whether residue C394 stabilizes $A_{2A}R$ dimerization by forming disulfide linkages, we incubated the SEC-separated dimers of $A_{2A}R$-WT and $A_{2A}R$-Q372ΔC with 5 mM of the reducing agent TCEP, followed by SDS-PAGE and western blotting. The population of each species was determined

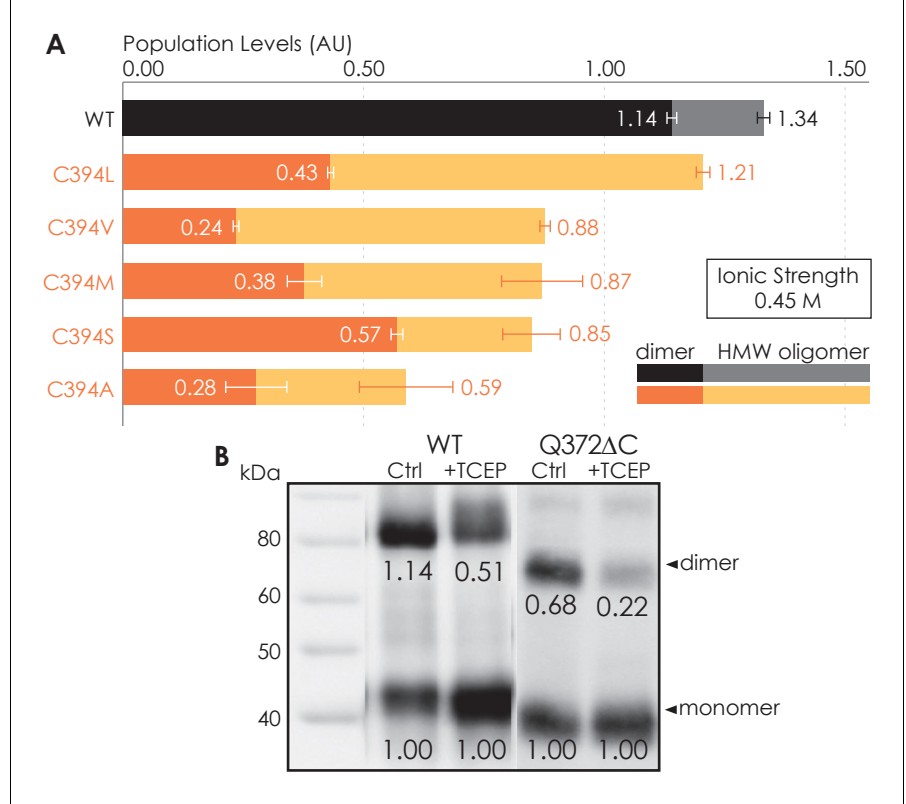

**Figure 2.** Residue C394 helps stabilize A$_{2A}$R oligomerization via disulfide bonds. (**A**) The effect of C394X substitutions on A$_{2A}$R oligomerization. The levels of dimer (dark colors) and high-molecular-weight oligomer (light colors) are expressed relative to the monomeric population in arbitrary units, with reported errors calculated from the variance of the fit, not experimental variation. (**B**) Line densitometry of western blot bands on size-exclusion chromatography (SEC)-separated dimeric populations of A$_{2A}$R-WT and Q372ΔC with and without 5 mM TCEP. The level of dimer is expressed relative to the monomeric population in arbitrary units similarly to the SEC analysis. MagicMark protein ladder (LC5602) is used as the molecular weight standard.

The online version of this article includes the following source data for figure 2:

**Source data 1.** Raw western blot of size-exclusion chromatography-separated dimeric populations of A$_{2A}$R-WT with and without 5 mM TCEP.

**Source data 2.** Raw western blot of size-exclusion chromatography-separated dimeric populations of A$_{2A}$R-WT with and without 5 mM TCEP.

**Source data 3.** Raw western blot of size-exclusion chromatography-separated dimeric populations of A$_{2A}$R-Q372ΔC with and without 5 mM TCEP.

**Source data 4.** Raw western blot of size-exclusion chromatography-separated dimeric populations of A$_{2A}$R-Q372ΔC with and without 5 mM TCEP.

**Source data 5.** Raw size-exclusion chromatography data of A$_{2A}$R-WT and C394X variants.

as the area under the densitometric trace. The dimer level was then expressed as monomer-equivalent concentration ratios in a manner similar to that of the SEC experiment described above. Upon incubation with TCEP, the dimer level of the A$_{2A}$R-WT sample decreased from 1.14 to 0.51 (*Figure 2B*). This indicates that disulfide bond formation via residue C394 is one possible mechanism for A$_{2A}$R dimerization. Interestingly, the dimer level of the A$_{2A}$R-Q372ΔC sample also decreased from 0.68 to 0.22 (*Figure 2B*). This suggests that there may exist other inter-A$_{2A}$R disulfide bonds that do not involve residue C394. Still, in both cases, a clearly visible population of A$_{2A}$R dimer persists, even after reduction of disulfide bonds via TCEP (*Figure 2B*), suggesting that there must be additional interfacial sites that help drive A$_{2A}$R dimer/oligomerization.

## C-terminus truncation systematically reduces $A_{2A}R$ oligomerization

To determine which interfacial sites in the C-terminus other than the disulfide-bonded cysteines drive $A_{2A}R$ dimer/oligomerization, we carried out systematic truncations at eight sites along the C-terminus (A316, V334, G344, G349, P354, N359, Q372, and P395), generating eight $A_{2A}R$-$\Delta C$ variants (*Figure 3A*). The $A_{2A}R$-A316$\Delta C$ variant corresponds to the removal of the entire disordered C-terminal region and is used in all published structural studies of $A_{2A}R$ (*Martynowycz et al., 2020*; *Song et al., 2020*; *García-Nafría et al., 2018*; *Sun et al., 2017*; *Carpenter et al., 2016*; *Hino et al., 2012*; *Xu et al., 2011*; *Lebon et al., 2011*; *Doré et al., 2011*; *Jaakola et al., 2008*; *Fanelli and Felline, 2011*). Using the SEC analysis described earlier (*Figure 1*), we evaluated the HMW oligomer and dimer levels of the $A_{2A}R$-$\Delta C$ variants relative to that of the $A_{2A}R$ full-length-wild-type (FL-WT) control. Both the dimer and the total oligomer levels of $A_{2A}R$ decreased progressively with the shortening of the C-terminus, with almost no oligomerization detected upon complete truncation of the C-terminus at site A316 (*Figure 3B*). This result shows that the C-terminus drives $A_{2A}R$ oligomerization, with multiple potential interaction sites positioned along much of its length.

Interestingly, there occurred a dramatic decrease in the dimer level between the N359 and P354 truncation sites, from a value of 0.81 to 0.19, respectively (*Figure 3B*). A similar result, though less pronounced, was observed on the total oligomer level, with a decrease from 1.09 to 0.62 for the N359 and P354 truncation sites, respectively (*Figure 3B*). Clearly, the C-terminal segment encompassing residues 354–359 (highlighted in black in *Figure 3A*) is a key constituent of the $A_{2A}R$ oligomeric interface.

Since segment 354–359 contains three consecutive charged residues ([355]ERR[357]; *Figure 3A*), which could be involved in electrostatic interactions, we hypothesized that this [355]ERR[357] cluster could strengthen inter-protomer $A_{2A}R$-$A_{2A}R$ association. To test this hypothesis, residues [355]ERR[357] were substituted by [355]AAA[357] on $A_{2A}R$-FL-WT and $A_{2A}R$-N359$\Delta C$ to generate $A_{2A}R$-ERR:AAA variants (*Figure 3C*). We then compared the HMW oligomer and dimer levels of the resulting variants with controls (same $A_{2A}R$ variants but without the ERR:AAA mutations). We found that the ERR:AAA mutations had varied effects on the dimer level: decreasing for $A_{2A}R$-FL-WT (ctrl: 0.49; ERR:AAA: 0.29) but increasing for $A_{2A}R$-N359$\Delta C$ (ctrl: 0.33; ERR:AAA: 0.48) (*Figure 3C*). In contrast, the ERR:AAA mutations reduced the HMW oligomer level of both $A_{2A}R$-FL-WT (ctrl: 0.88; ERR:AAA: 0.66) and $A_{2A}R$-N359$\Delta C$ (ctrl: 0.68; ERR:AAA: 0.38) (*Figure 3C*). Consistently, the ERR:AAA mutation lowered the total oligomer level of both $A_{2A}R$-FL-WT (ctrl: 1.37; ERR:AAA: 0.94) and $A_{2A}R$-N359$\Delta C$ (ctrl: 1.01; ERR:AAA: 0.85) (*Figure 3C*). These results suggest that the charged residues [355]ERR[357] participate in $A_{2A}R$ oligomerization, with a greater effect in the context of a longer C-terminus and for forming higher-order oligomers. The question then arises as to what types of interactions are formed along the C-terminus that help stabilize $A_{2A}R$ oligomerization.

## C-terminus truncation disrupts complex network of non-bonded interactions necessary for $A_{2A}R$ dimerization

Given that the structure of $A_{2A}R$ dimers or oligomers is unknown, we next used MD simulations to seek molecular-level insights into the role of the C-terminus in driving $A_{2A}R$ dimerization and to gain an understanding of what types of interactions and sites may be involved in this process. First, to explore $A_{2A}R$ dimeric interface, we performed coarse-grained (CG) MD simulations using the Martini force field (see Materials and methods for details). The Martini force field can access the length and time scales relevant to membrane protein oligomerization, albeit at the expense of atomic-level details. We carried out a series of CGMD simulations on five $A_{2A}R$-$\Delta C$ variants designed to mirror the experiments by systematic truncation at five sites along the C-terminus (A316, V334, P354, N359, and C394). Our results revealed that $A_{2A}R$ dimers were formed with multiple interfaces, all involving the C-terminus only (*Figure 4A*). The transmembrane heptahelical bundles were not a part of the dimeric interfaces as they all showed distances greater than the minimum distance criterion of 7 Å for interacting helices. The vast majority of $A_{2A}R$ dimers were symmetric, with the C-termini of the protomers directly interacting with each other. A smaller fraction of the dimers had asymmetric orientations, with the C-terminus of one protomer interacting with other parts of the other protomer, such as ICL2 (the second intracellular loop) and ICL3 (*Figure 4A*).

Our observation of multiple $A_{2A}R$ oligomeric interfaces, which is consistent with previous studies (*Fanelli and Felline, 2011*; *Song et al., 2020*), suggests that tunable, non-covalent

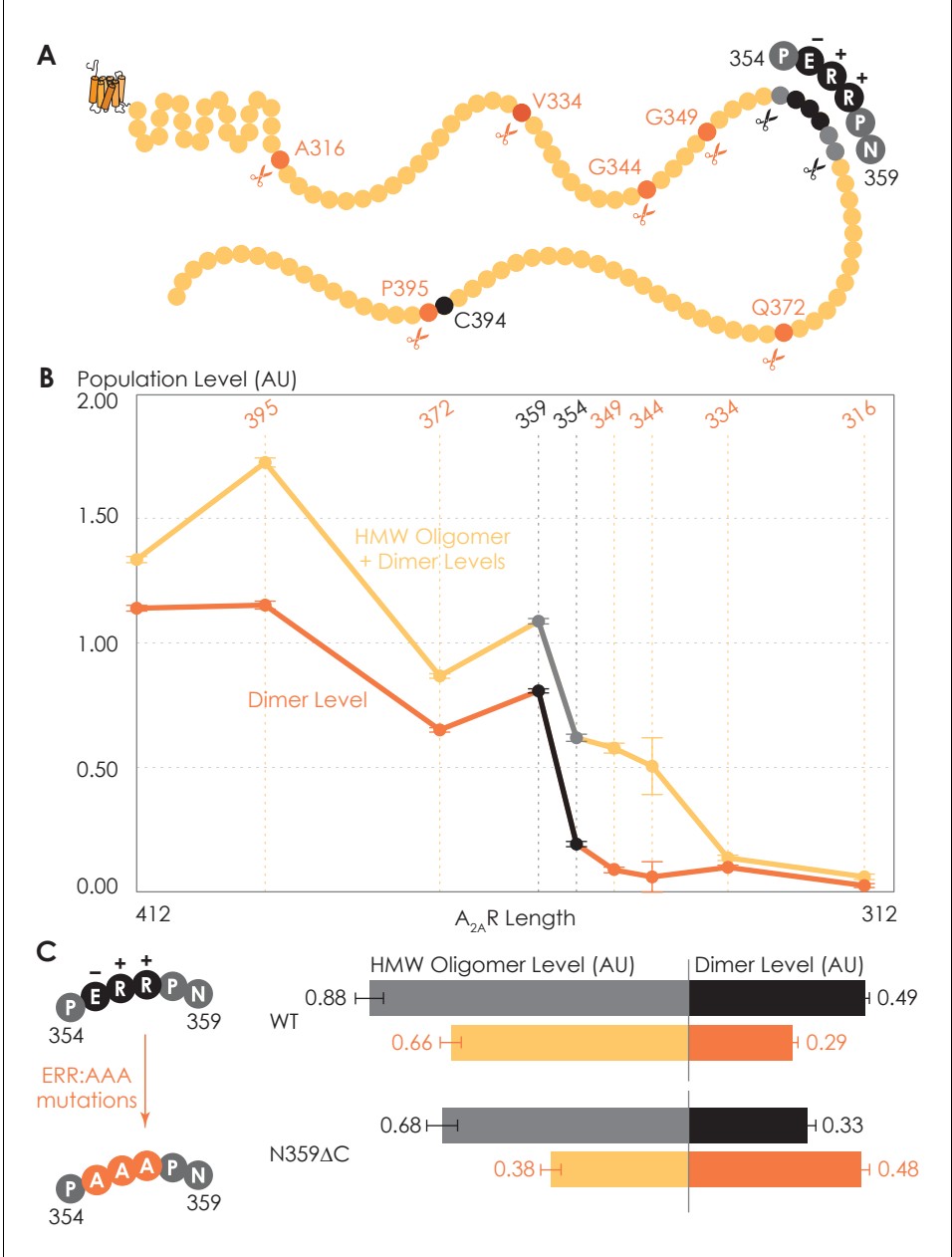

**Figure 3.** Truncating the C-terminus systematically affects A₂ₐR oligomerization. (**A**) Depiction of where the truncation points are located on the C-terminus, with region 354–359 highlighted (in black) showing critical residues. (**B**) The levels of dimer and high-molecular-weight (HMW) oligomer are expressed relative to the monomeric population as an arbitrary unit and plotted against the residue number of the truncation sites, with reported errors calculated from the variance of the fit, not experimental variation. Region 354–359 is emphasized (in black and gray) due to a drastic change in the dimer and HMW oligomer levels. (**C**) The dependence of A₂ₐR oligomerization on three consecutive charged residues $^{355}$ERR$^{357}$. The substitution of residues $^{355}$ERR$^{357}$ to $^{355}$AAA$^{357}$ is referred to as the ERR:AAA mutations. The levels of dimer and HMW oligomer are expressed relative to the monomeric population as an arbitrary unit, with reported errors calculated from the variance of the fit, not experimental variation.

The online version of this article includes the following source data for figure 3:

**Source data 1.** Raw size-exclusion chromatography data of A₂ₐR-WT and C-terminally truncated ΔC variants.

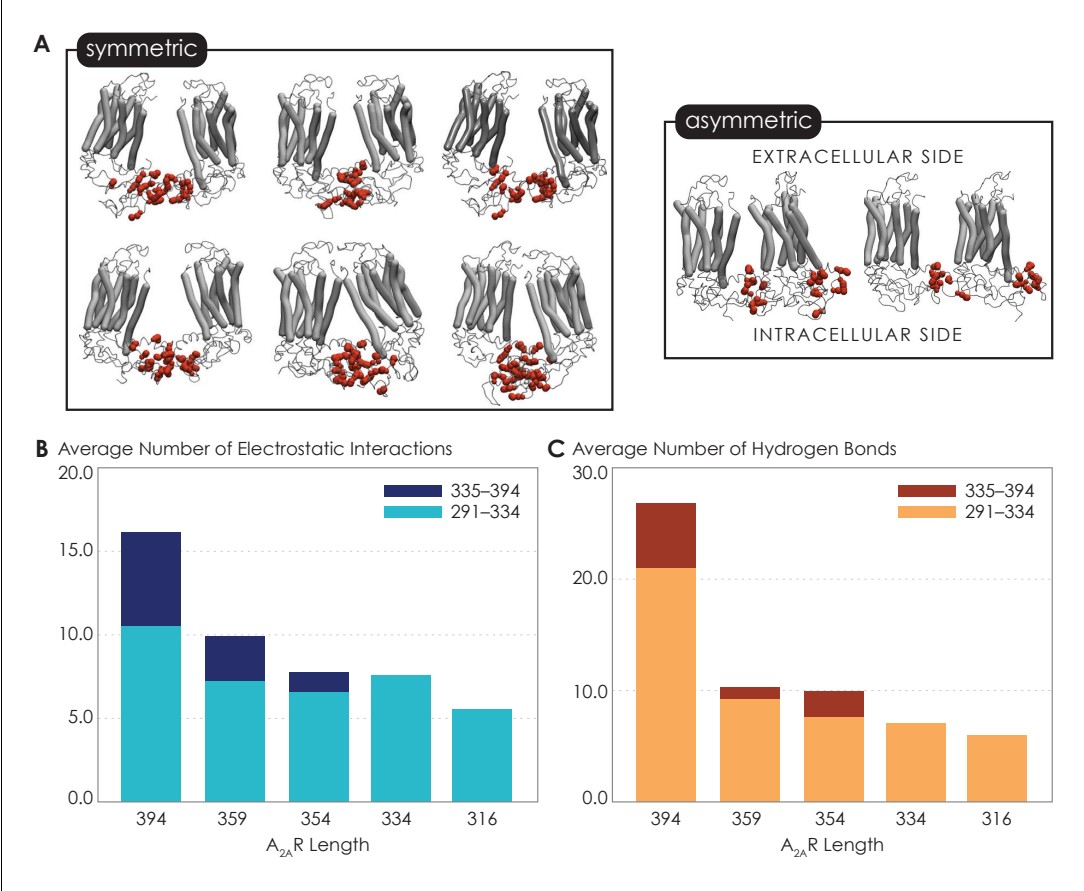

**Figure 4.** Non-bonded interactions of the extended C-terminus of A$_{2A}$R play a critical role in stabilization of the dimeric interface. (**A**) Dimer configurations from cluster analysis in GROMACS of the 394-residue variant identify two major clusters involving either (1) the C-terminus of one protomer and the C-terminus, ICL2, and ICL3 of the second protomer or (2) the C-terminus of one protomer and ICL2, ICL3, and ECL2 of the second protomer. Spheres: residues forming intermolecular electrostatic contacts. (**B**) Average number of residues that form electrostatic contacts as a function of sequence length of A$_{2A}$R. (**C**) Average number of residues that form hydrogen bonds as a function of sequence length of A$_{2A}$R. The criteria for designating inter-A$_{2A}$R contacts as electrostatic interactions or hydrogen bonds are described in detail in Materials and methods.

The online version of this article includes the following source data for figure 4:

**Source data 1.** Detailed data regarding the multiple interfaces of A$_{2A}$R and the network of non-bonded interactions that stabilize these interfaces.

intermolecular interactions may be involved in receptor dimerization. We first dissected two key non-covalent interaction types: electrostatic and hydrogen bonding interactions. Electrostatic interactions were calculated from CGMD simulations, while hydrogen bonds were quantified from atomistic MD simulation as the CG model merges all hydrogens into a CG bead and hence cannot report on hydrogen bonds. This analysis was performed on the symmetric dimers as they constituted the more dominant population. With the least truncated A$_{2A}$R variant containing the longest C-terminus, A$_{2A}$R-C394ΔC, we observed an average of 15.9 electrostatic contacts (*Figure 4B*) and 26.7 hydrogen bonds (*Figure 4C*) between the C-termini of the protomers. This result shows that both electrostatic interactions and hydrogen bonds can play important roles in A$_{2A}$R dimer formation.

Upon further C-terminus truncation, the average number of both electrostatic contacts and hydrogen bonds involving C-terminal residues progressively declined, respectively reaching 5.4 and 6.0 for A$_{2A}$R-A316ΔC (in which the disordered region of the C-terminus is removed) (*Figure 4B, C*). This result is consistent with the experimental result, which demonstrated a progressive decrease of A$_{2A}$R oligomerization with the shortening of the C-terminus (*Figure 3B*). Interestingly, upon systematic truncation of the C-terminal segment 335–394, we observed in segment 291–334 a steady decrease in the average number of electrostatic contacts, from 10.4 to 7.4 (*Figure 4B*). This trend was even more pronounced with hydrogen bonding contacts involving segment 291–334 decreasing

drastically from 21.0 to 7.0 as segment 335–394 was gradually removed (*Figure 4C*). This observation that truncation of a C-terminal segment reduces inter-A$_{2A}$R contacts elsewhere along the C-terminus indicates that an allosteric mechanism of dimerization exists, in which an extended C-terminus of A$_{2A}$R stabilizes inter-A$_{2A}$R interactions near the heptahelical bundles of the dimeric complex. These results demonstrate that A$_{2A}$R dimers can be formed via multiple interfaces and stabilized by an allosteric network of electrostatic interactions and hydrogen bonds along much of its C-terminus.

## Ionic strength modulates oligomerization of C-terminally truncated A$_{2A}$R variants

So far, we have demonstrated that the C-terminus clearly plays a role in forming A$_{2A}$R oligomeric interfaces. However, it remains unclear what the driving factors of A$_{2A}$R oligomerization are and whether the oligomeric populations are thermodynamic products. The variable nature of A$_{2A}$R oligomeric interfaces suggests that the main driving forces must be non-covalent interactions, such as electrostatic interactions and hydrogen bonds. Modulating the solvent ionic strength is an effective method to identify the types of non-covalent interaction(s) at play. Specifically, with increasing ionic strength, electrostatic interactions are weakened (based on Debye–Hückel theory, most electrostatic bonds at a distance greater than 5 Å are screened out at an ionic strength of 0.34 M at 4°C) and depletion interactions are enhanced with salting-out salts, while hydrogen bonds remain relatively impervious. For this reason, we subjected various A$_{2A}$R variants (FL-WT, FL-ERR:AAA, N359ΔC, and V334ΔC) to ionic strength ranging from 0.15 to 0.95 M by adding NaCl (buffer composition shown in Materials and methods). The HMW oligomer and dimer levels of the four A$_{2A}$R variants were determined and plotted as a function of ionic strengths.

The low ionic strength of 0.15 M should not affect hydrogen bonds or electrostatic interactions if present. We found that the dimer and total oligomer levels of all four variants were near zero (*Figure 5*). This is a striking experimental observation: despite being shown to play a role in stabilizing A$_{2A}$R dimers according to our MD simulations (*Figure 4B, C*), we can conclude that electrostatic and hydrogen-bonding interactions are not the dominant driving force for A$_{2A}$R association. The question remains whether depletion interactions could facilitate A$_{2A}$R oligomerization.

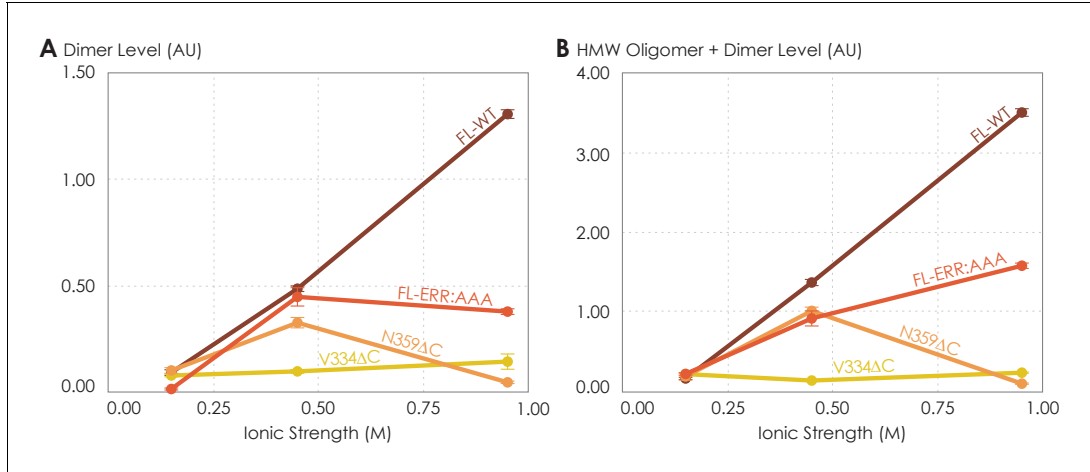

**Figure 5.** The effects of ionic strength on the oligomerization of various A$_{2A}$R variants reveal the involvement of depletion interactions. The levels of (**A**) dimer and (**B**) high-molecular-weight oligomer + dimer are expressed relative to the monomeric population as an arbitrary unit and plotted against ionic strength, with reported errors calculated from the variance of the fit, not experimental variation. NaCl concentration is varied to achieve ionic strengths of 0.15, 0.45, and 0.95 M.

The online version of this article includes the following source data and figure supplement(s) for figure 5:

**Source data 1.** Raw size-exclusion chromatography data of various A$_{2A}$R variants under different ionic strengths of 0.15, 0.45, and 0.95 M.

**Figure supplement 1.** The dimer/oligomerization of A$_{2A}$R is a thermodynamic process where the dimer and high-molecular-weight oligomer once formed are kinetically trapped.

**Figure supplement 1—source data 1.** Raw size-exclusion chromatography (SEC) data of the consecutive rounds of SEC performed on A$_{2A}$R-WT and Q372ΔC.

At higher ionic strengths of 0.45 M and 0.95 M, the dimer and total oligomer levels of $A_{2A}R$-V334ΔC still remained near zero (*Figure 5*). In contrast, we observed a progressive and significant increase in the dimer and total oligomer levels of $A_{2A}R$-FL-WT with increasing ionic strength (*Figure 5*). This result indicates that $A_{2A}R$ oligomerization is driven by depletion interactions enhanced with increasing ionic strength and that these interactions must involve the C-terminal segment after residue V334.

Upon closer examination, we recognize that at the very high ionic strength of 0.95 M the increase in the dimer and total oligomer levels was robust for $A_{2A}R$-FL-WT, but less pronounced for $A_{2A}R$-FL-ERR:AAA (*Figure 5*). Furthermore, this high ionic strength even had an opposite effect on $A_{2A}R$-N359ΔC, with both its dimer and total oligomer levels abolished (*Figure 5*). These results indicate that the charged cluster [355]ERR[357] and the C-terminal segment after residue N359 promote the depletion interactions to drive $A_{2A}R$ oligomerization. Taken together, we can conclude that $A_{2A}R$ oligomerization is more robust when the C-terminus is fully present and the ionic strength higher, suggesting that depletion interactions via the C-terminus are strong driving factors of $A_{2A}R$ oligomerization.

The discussion of depletion interactions as driving factors assumes that $A_{2A}R$ dimer/oligomer populations are thermodynamic products at equilibrium with the $A_{2A}R$ monomer population. However, some of the $A_{2A}R$ dimer/oligomer populations may be kinetically stabilized. To address this question, we tested the stability and reversibility of $A_{2A}R$ oligomers by performing a second round of SEC on the monomer and dimer/oligomer populations of the $A_{2A}R$-WT and Q372ΔC variants. We found that the SEC-separated monomers repopulate into dimer/oligomer, with the total oligomer level after redistribution comparable with that of the initial samples for both $A_{2A}R$-WT (initial: 2.87; redistributed: 1.60) and Q372ΔC (initial: 1.49; redistributed: 1.40) (*Figure 5—figure supplement 1A*). This observation indicates that $A_{2A}R$ oligomer is a thermodynamic product with a lower free energy compared with that of the monomer (*Figure 5—figure supplement 1B*). This agrees with the results we have shown in *Supplementary file 1* that the oligomer levels of $A_{2A}R$-WT are consistent among replicates (1.34–2.05) and that $A_{2A}R$ oligomerization can be modulated with ionic strengths via depletion interactions (*Figure 5*).

In contrast, the SEC-separated dimer/oligomer populations do not repopulate to form monomers (*Figure 5—figure supplement 1A*). This observation is consistent with a published study of ours on $A_{2A}R$ dimers (*Schonenbach et al., 2016*), indicating that once the oligomers are formed, some are kinetically trapped and thus cannot redistribute into monomers. We believe that disulfide linkages are likely candidates to kinetically stabilize $A_{2A}R$ oligomers, as demonstrated by their redistribution into monomers only in the presence of a reducing agent (*Figure 2B*).

Taken together, we suggest that $A_{2A}R$ oligomerization is a thermodynamic process (*Figure 5—figure supplement 1B*), with the free energy of the dimer/oligomers lowered by depletion forces that hence increase their population relative to that of the monomers (there always exists a distribution between the two). Once formed, the redistributed dimer/oligomer populations may be kinetically stabilized by disulfide linkages. The question then arises whether inter-$A_{2A}R$ interactions are primarily a result of the C-termini directly interacting with one another. This question motivated us to carry out a study focused on investigating the behavior of $A_{2A}R$ C-terminus sans the transmembrane domains.

## The isolated $A_{2A}R$ C-terminus is prone to aggregation

To test whether $A_{2A}R$ oligomerization is driven by direct depletion interactions among the C-termini of the protomers, we assayed the solubility and assembly properties of the stand-alone $A_{2A}R$ C-terminus—an intrinsically disordered peptide—sans the upstream transmembrane regions. Since depletion interactions can be manifested via the hydrophobic effect (*van der Vegt et al., 2017*), we examined whether this effect can also drive the assembly of the $A_{2A}R$ C-terminal peptides.

It is an active debate whether the hydrophobic effect can be promoted or suppressed by ions with salting-out or salting-in tendency, respectively (*Thomas and Elcock, 2007*; *Graziano, 2010*; *Zangi et al., 2007*; *Grover and Ryall, 2005*). We increased the solvent ionic strength using either sodium (salting-out) or guanidinium (salting-in) ions and assessed the aggregation propensity of the C-terminal peptides using UV-Vis absorption at 450 nm, which indicates the turbidity of the solution. We first observed the behavior of the C-terminus with increasing salting-out NaCl concentrations. At NaCl concentrations below 1 M, the peptide was dominantly soluble, despite showing slight

aggregation at NaCl concentrations between 250 and 500 mM (**Figure 6A**). At NaCl concentrations above 1 M, $A_{2A}R$ C-terminal peptides strongly associated into insoluble aggregates (**Figure 6A**). Consistent with the observations made with the intact receptor (**Figure 5**), the $A_{2A}R$ C-terminus showed the tendency to progressively associate and eventually precipitate with increasing ionic strengths, suggesting that depletion interactions drive the association and precipitation of the peptides. We next observed the behavior of the C-terminus with increasing concentrations of guanidine hydrochloride (GdnHCl), which contains salting-in cations that do not induce precipitation and instead facilitate the solubilization of proteins (**Heyda et al., 2017**; **Baldwin, 1996**). Our results demonstrated that the $A_{2A}R$ C-terminus incubated in 4 M GdnHCl showed no aggregation propensity (**Figure 6A**), validating our expectation that salting-in salts do not enhance depletion interactions. These observations demonstrate that the C-terminal peptide in and of itself, outside the context of the lipid membrane and TM domain, can directly interact with other C-terminal peptides to form self-aggregates in the presence of ions, and presumably solutes, that have salting-out effects.

Attractive hydrophobic interactions among the hydrophobic residues are further enhanced when the water that solvate the protein surface have more favorable interactions with other water molecules, ions, or solutes than with the protein surface, here the truncated C-terminus (**Larsen et al., 1998**; **Tsai and Nussinov, 1997**; **Tsai et al., 1997**). We explored the possible contribution of hydrophobic interactions to the aggregation of the C-terminal peptides using both experimental and computational approaches. Using differential scanning fluorimetry (DSF), we gradually increased the temperature to melt the C-terminal peptides, exposing any previously buried hydrophobic residues (**Figure 6—figure supplement 1A, B**), which then bound to the SYPRO orange fluorophore, resulting in an increase in fluorescence signal. Our results showed that as the temperature increased, a steady rise in fluorescence was observed (**Figure 6B**), indicating that multiple hydrophobic residues were gradually exposed to the SYPRO dye. However, at approximately 65°C, the melt peak signal was abruptly quenched (**Figure 6B**), indicating that the hydrophobic residues were no longer exposed to the dye. This observation suggests that, at 65°C, enough hydrophobic residues in the C-terminal peptides become exposed such that they collapse on one another (thus expelling the bound dye molecules), resulting in aggregation. This experimental result is further supported by our CGMD computational analysis of C-terminal non-polar contacts found in $A_{2A}R$ symmetrical dimers (**Figure 6—figure supplement 1C**). Specifically, we observed an average of 60 non-polar contacts for $A_{2A}R$-C394ΔC. This number progressively declined upon further C-terminus truncation, reaching

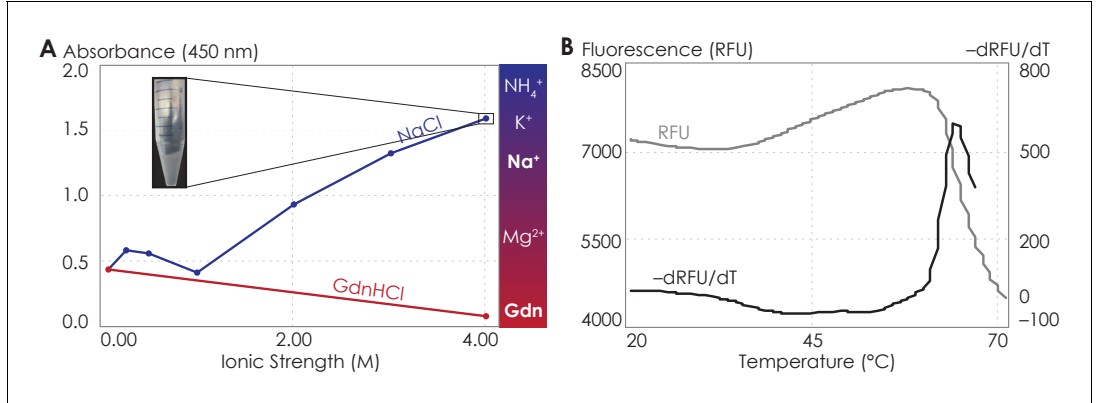

**Figure 6.** The $A_{2A}R$ C-terminus is prone to aggregation. (**A**) Absorbance at 450 nm of the $A_{2A}R$ C-terminus in solution, with NaCl and GdnHCl concentrations varied to achieve ionic strengths 0–4 M. Inset: the solution at ionic strength 4 M achieved with NaCl. The Hofmeister series is provided to show the ability of cations to salt-out (blue) or salt-in (red) proteins. (**B**) SYPRO orange fluorescence of solutions containing the $A_{2A}R$ C-terminus as the temperature was varied from 20°C to 70°C (gray). The change in fluorescence, measured in relative fluorescence unit (RFU), was calculated by taking the first derivative of the fluorescence curve (black).

The online version of this article includes the following source data and figure supplement(s) for figure 6:

**Source data 1.** Detailed data showing the propensity of $A_{2A}R$ C-terminus to aggregate.

**Figure supplement 1.** The C-terminus of $A_{2A}R$ can form non-polar contacts.

**Figure supplement 1—source data 1.** Detailed data showing the ability of $A_{2A}R$ C-terminus to form non-polar contacts.

15 for $A_{2A}$R-A316ΔC. Clearly, the hydrophobic effect can cause $A_{2A}$R C-terminal peptides to directly associate. These results demonstrate that $A_{2A}$R oligomer formation can be driven by depletion interactions among the C-termini of the protomers by non-polar contacts.

## Discussion

The key finding of this study is that the C-terminus of $A_{2A}$R, removed in all previously published structural studies, is directly responsible for receptor oligomerization. Using a combination of experimental and computational approaches, we demonstrate that the C-terminus stabilizes $A_{2A}$R oligomers via a combination of disulfide linkages, hydrogen bonds, electrostatic interactions, and hydrophobic interactions. This diverse combination of interactions is greatly enhanced by depletion interactions, forming a network of malleable bonds that drives $A_{2A}$R oligomerization and gives rise to multiple oligomeric interfaces.

Intermolecular disulfide linkages play a role in $A_{2A}$R oligomerization, potentially by kinetically trapping the receptor oligomers. Among the seven cysteines that do not form intramolecular disulfide bonds (*De Filippo et al., 2016*; *Naranjo et al., 2015*; *O'Malley et al., 2010*), residue C394 is largely involved in stabilizing $A_{2A}$R oligomers (*Figure 2A*). Indeed, this cysteine is highly conserved and a C-terminal cysteine is almost always present in $A_{2A}$R homologs (*Pándy-Szekeres et al., 2018*), suggesting that it may serve an important role in vivo. There may also exist inter-$A_{2A}$R disulfide linkages that do not involve residue C394 at all as the SEC-separated dimer/oligomer populations of $A_{2A}$R-Q372ΔC, which lack residue C394, were still resistant to TCEP reduction (*Figure 2B*) and appear to be kinetically trapped (*Figure 5—figure supplement 1*). Such disulfide linkages may involve other cysteines in the hydrophobic core of $A_{2A}$R, namely $C28^{1.54}$, $C82^{3.30}$, $C128^{4.49}$, $C185^{5.46}$, $C245^{6.47}$, or $C254^{6.56}$. Many examples exist where disulfide linkages help drive GPCR oligomerization, including the CaR-mGluR$_1$ heterodimer (*Gama et al., 2001*), homodimers of mGluR$_5$ (*Romano et al., 1996*), M$_3$R (*Zeng and Wess, 1999*), V$_2$R (*Zhu and Wess, 1998*), 5-HT$_4$R (*Berthouze et al., 2007*) and 5-HT$_{1D}$R (*Lee et al., 2000*), and even higher-order oligomers of D$_2$R (*Guo et al., 2008*). Although unconventional cytoplasmic disulfide bonds have been reported (*Saaranen and Ruddock, 2013*; *Locker and Griffiths, 1999*), no study has shown how such linkages would be formed in vivo as the cytoplasm lacks the conditions and machinery required for disulfide bond formation (*Gaut and Hendershot, 1993*; *Hwang et al., 1992*; *Helenius et al., 1992*; *Creighton et al., 1980*).

The electrostatic interactions that stabilize $A_{2A}$R oligomer formation come from multiple sites along the C-terminus. From a representative snapshot of a $A_{2A}$R-C394ΔC dimer from our MD simulations (*Figure 7A*), we could visualize not only the intermolecular interactions calculated from the CGMD simulations (*Figure 4B*), but also intramolecular salt bridges. In particular, the $^{355}$ERR$^{357}$ cluster of charged residues lies distal from the dimeric interface but still forms several salt bridges (*Figure 7A*, inset). This observation is supported by our experimental results showing that substituting this charged cluster with alanines reduces the total $A_{2A}$R oligomer levels (*Figure 3C*). However, it is unclear how such salt bridges involving this $^{355}$ERR$^{357}$ cluster are enhanced by depletion interactions (*Figure 5*) as electrostatic interactions are usually screened out at high ionic strengths. In our MD simulations, we also observed networks of salt bridges along the dimeric interface, for example, between K315 of one monomer and D382 and E384 of the other monomer (*Figure 7A*, inset). The innate flexibility of the C-terminus could facilitate the formation of such salt bridges, which then help stabilize $A_{2A}$R dimers.

Our finding that $A_{2A}$R forms homo-oligomers via multiple interfaces (*Figure 4A*) agrees with the increasing number of studies reporting multiple and interconverting oligomeric interfaces in $A_{2A}$R and other GPCRs (*Song et al., 2020*; *Ghosh et al., 2014*; *Periole et al., 2012*; *Fanelli and Felline, 2011*; *Liu et al., 2012*; *Huang et al., 2013*; *Manglik et al., 2012*; *Thorsen et al., 2014*; *Fotiadis et al., 2006*; *Fotiadis et al., 2003*; *Liang et al., 2003*; *Xue et al., 2015*; *Dijkman et al., 2018*). When translated to in vivo situations, GPCR oligomers can also transiently associate and dissociate (*Kasai et al., 2018*; *Tabor et al., 2016*; *Möller et al., 2020*; *Vilardaga et al., 2008*). Such conformational changes require that the oligomeric interfaces be formed by interactions that can easily be modulated. This is consistent with our study, which demonstrates that depletion interactions via the intrinsically disordered, malleable C-terminus drive $A_{2A}$R oligomerization. Because depletion interactions can be readily tuned by environmental factors, such as ionic strength,

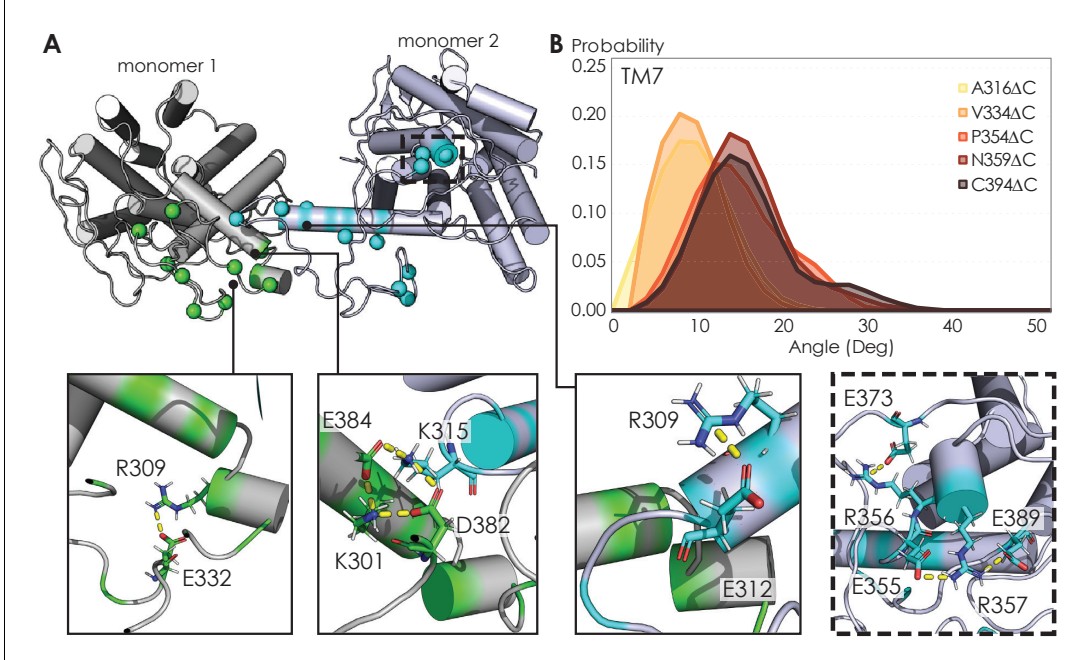

**Figure 7.** Visualizing A2AR dimeric interface and observing conformational changes of the TM7 using MD simulations. **(A)** Representative snapshot of $A_{2A}R$-C394ΔC dimers shows salt bridge formation between a sample trajectory. The insets are closeups of the salt bridges, which can be both intra- and intermolecular. The last inset shows a network of salt bridges with the charged cluster [355]ERR[357] involved. **(B)** Helical tilt angles for TM7 helix in $A_{2A}R$ as a function of protein length. Systematic truncations of the C-terminus lead to rearrangement of the heptahelical bundle. The participation of the C-terminus in $A_{2A}R$ dimerization increases the tilting of the TM7 domain, which is in closest proximity to the C-terminus.

The online version of this article includes the following source data and figure supplement(s) for figure 7:

**Source data 1.** MD simulations data used to visualize $A_{2A}R$ dimeric interface and observe the conformational changes of the TM7.

**Figure supplement 1.** Helical tilt angles for TM1–6 helices in $A_{2A}R$ as a function of protein length.

**Figure supplement 1—source data 1.** Helical tilt angles for TM1–6 helices in $A_{2A}R$ as a function of protein length.

molecular crowding, and temperature, the formation of GPCR oligomeric complexes could be dynamically modulated in response to environmental cues to regulate receptor function.

Not only did we find multiple $A_{2A}R$ oligomeric interfaces, we also found that these interfaces can be either symmetric or asymmetric. This finding is supported by a growing body of evidence that there exists both symmetric and asymmetric oligomeric interfaces for $A_{2A}R$ (*Song et al., 2020*) and many other GPCRs. Studies using various biochemical and biophysical techniques have shown that heterotetrameric GPCR complexes can be formed by dimers of dimers, including µOR-δ OR (*Golebiewska et al., 2011*), CXC$_4$R-CC$_2$R (*Armando et al., 2014*), CB$_1$R/D$_2$R (*Bagher et al., 2017*), as well as those involving $A_{2A}R$, such as A$_1$R-A$_{2A}R$ (*Navarro et al., 2018a*; *Navarro et al., 2016*) and $A_{2A}R$-D$_2$R (*Navarro et al., 2018b*). The quaternary structures identified in these studies required specific orientations of each protomer, with the most viable model involving a stagger of homodimers with symmetric interfaces (*DelaCuesta-Barrutia et al., 2020*). On the other hand, since symmetric interfaces limit the degree of receptor association to dimers, the HMW oligomer of $A_{2A}R$ observed in this (*Song et al., 2020*) and other studies (*Schonenbach et al., 2016*; *Vidi et al., 2008*) can only be formed via asymmetric interfaces. It is indeed tempting to suggest that the formation of the HMW oligomer of $A_{2A}R$ may even arise from combinations of different interfaces. In any case, the wide variation of GPCR oligomerization requires the existence of both symmetric and asymmetric oligomeric interfaces.

The ultimate question to answer is how oligomerization alters $A_{2A}R$ function. In the case of $A_{2A}R$, displacement of the transmembrane domains has been demonstrated to be the hallmark of receptor activation (*Eddy et al., 2018*; *Sušac et al., 2018*; *Prosser et al., 2017*; *Ye et al., 2016*), but no studies have linked receptor oligomerization with the arrangement of the TM bundles in $A_{2A}R$. Our MD simulations revealed that C-terminus truncation resulted in structural changes in the heptahelical

bundles of $A_{2A}R$ dimers. Specifically, as more of the C-terminus was preserved, we observed a progressive increase in the helical tilt of TM7 (*Figure 7B*). This change in helical tilt occurred for the entire heptahelical bundle, with an increase in tilt for TM1, TM2, TM3, TM5, and TM7, and a decrease in tilt for TM4 and TM6 (*Figure 7—figure supplement 1*). The longer C-terminus in the full-length $A_{2A}R$ permits greater rearrangements in the transmembrane regions, leading to the observed change in helical tilt. Furthermore, in the cellular context, it has been demonstrated that truncation of the C-terminus significantly reduced receptor association with $G\alpha_s$ and cAMP production in cellular assays (*Koretz et al., 2021*). These results hint at potential conformational changes of $A_{2A}R$ upon oligomerization, necessitating future investigation on functional consequences.

Like all biophysical studies of membrane proteins in non-native environments, a drawback in our study is the question whether the above results, conducted in detergent micelles, can be translated to bilayer or cellular context. It has been demonstrated that the propensity of membrane proteins to associate and oligomerize is greater in lipid bilayers compared to that in detergent micelles (*Popot and Engelman, 1990*). Furthermore, in the cellular context, $A_{2A}R$ has been shown to assemble into homo-oligomers in transfected HEK293 cells (*Canals et al., 2003*) and in Cath.A differentiated neuronal cells (*Vidi et al., 2008*), while C-terminally truncated $A_{2A}R$ shows no protein aggregation or clustering on the cell surface, in contrast with its WT form (*Burgueño et al., 2003*). Therefore, we speculate that $A_{2A}R$ oligomerization will be present in the lipid bilayer and cellular environment. Regardless, given that most biophysical structure-function studies of GPCRs are conducted in detergent micelles and other artificial membrane mimetics, it is critical to understand the role of the C-terminus in the oligomerization of $A_{2A}R$ reconstituted in detergent micelles.

C-terminal truncations prior to crystallization and structural studies may be the main reason for the scarcity of GPCR structures featuring oligomers. In that context, this study offers valuable insights and approaches into how the oligomerization of $A_{2A}R$ and potentially of other GPCRs can be tuned by modifying the intrinsically disordered C-terminus and varying salt types and concentrations. The presence of $A_{2A}R$ oligomeric populations with partial C-terminal truncations means that one can now study its oligomerization with less perturbation from the C-terminus. We also present evidence that the multiple C-terminal interactions that drive $A_{2A}R$ oligomerization can be easily modulated by ionic strength and specific salts (*Figures 5* and *6*). Given that ~75% and ~15% of all class A GPCRs possess a C-terminus of >50 and >100 amino acid residues (*Mirzadegan et al., 2003*), respectively, it will be worthwhile to explore the prospect of tuning GPCR oligomerization not only by shortening the C-terminus but also with simpler approaches such as modulating ionic strength and the surrounding salt environment.

## Conclusion

This study emphasizes for the first time the definite impact of the C-terminus on $A_{2A}R$ oligomerization, which can be extended to include the oligomers formed by other GPCRs with a protracted C-terminus. We have shown that the oligomerization of $A_{2A}R$ is strongly driven by depletion interactions along the C-terminus, further modulating and enhancing the multiple interfaces formed via a combination of hydrogen, electrostatic, hydrophobic, and covalent disulfide interactions. The task remains to link $A_{2A}R$ oligomerization to functional roles of the receptor. From a structural biology standpoint, visualizing the multiple oligomeric interfaces of $A_{2A}R$ in the presence of the full-length C-terminus is key to investigating whether these interfaces give rise to different oligomer functions.

## Materials and methods

**Key resources table**

| Reagent type (species) or resource | Designation | Source or reference | Identifiers | Additional information |
|---|---|---|---|---|
| Recombinant DNA reagent | pITy (plasmid) | *Parekh et al., 1996* | | |

*Continued on next page*

*Continued*

| Reagent type (species) or resource | Designation | Source or reference | Identifiers | Additional information |
|---|---|---|---|---|
| Strain, strain background (*Saccharomyces cerevisiae*) | BJ5464 | Robinson Lab – Carnegie Mellon University | | |
| Strain, strain background (*Escherichia coli*) | BL21 (DE3) | Sigma, St. Louis, MO, USA | #CMC0014 | |
| Chemical compound, drug | DDM | Anatrace, Maumee, OH, USA | #D310 | |
| Chemical compound, drug | CHAPS | Anatrace, Maumee, OH, USA | #C216 | |
| Chemical compound, drug | CHS | Anatrace, Maumee, OH, USA | #CH210 | |
| Chemical compound, drug | Xanthine amine congener | Sigma, St. Louis, MO, USA | #X103 | |
| Chemical compound, drug | Theophylline | Sigma, St. Louis, MO, USA | #T1633 | |
| Commercial assay, kit | Affigel 10 resin | BioRad, Hercules, CA, USA | #1536099 | |
| Commercial assay, kit | Tricorn Superdex 200 10/300 GL column | GE Healthcare, Pittsburgh, PA, USA | #17-5175-01 | |
| Antibody | Anti-$A_{2A}$R, clone 7F6-G5-A2 (Mouse monoclonal) | Millipore, Burlington, MA, USA | #05-717 | (1:500) dilution |
| Antibody | Anti-Mouse IgG H&L DyLight 550 (Goat monoclonal) | Abcam, Cambridge, MA, USA | #ab96880 | (1:600) dilution |
| Software, algorithm | MODELLER 9.23 | *Eswar et al., 2006* | | |
| Software, algorithm | martinize.py script | *de Jong et al., 2013* | | |
| Software, algorithm | ELNeDyn elastic network | *Periole et al., 2009* | | |
| Software, algorithm | MARTINI coarse-grained force field v2.2 | *Monticelli et al., 2008* | | |
| Software, algorithm | GROMACS 2016 | *Abraham et al., 2015* | | |
| Software, algorithm | backward.py script | *Wassenaar et al., 2014* | | |
| Software, algorithm | LINCS | *Hess et al., 1997* | | |
| Software, algorithm | CHARMM36 and TIP3P force fields | *Best et al., 2012*; *Jorgensen et al., 1983* | | |
| Software, algorithm | LOOS | *Romo and Grossfield, 2009* | | |
| Software, algorithm | VMD | *Humphrey et al., 1996* | | |

## Cloning, gene expression, and protein purification

The multi-integrating pITy plasmid (*Parekh et al., 1996*), previously used for overexpression of $A_{2A}$R in *Saccharomyces cerevisiae* (*O'Malley et al., 2009*), was employed in this study. pITy contains a Gal1–10 promoter for galactose-induced expression, a synthetic pre-pro leader sequence that

directs protein trafficking (*Clements et al., 1991*; *Parekh et al., 1995*), and the yeast alpha terminator. The genes encoding A$_{2A}$R variants with 10-His C-terminal tag were cloned into pITy downstream of the pre-pro leader sequence using either splice overlapping extension (*Bryksin and Matsumura, 2010*) or USER cloning using X7 polymerase (*Nørholm, 2010*; *Nour-Eldin et al., 2006*). The plasmids were then transformed into *S. cerevisiae* strain BJ5464 (MATα ura3-52 trp1 leu2Δ1 his3Δ200 pep4::HIS3 prb1Δ1.6R can1 GAL) (provided by the lab of Anne Robinson at Carnegie Mellon University) using the lithium-acetate/PEG method (*Gietz, 2014*). Transformants were selected on YPD G-418 plates (1% yeast extract, 2% peptone, 2% dextrose, 2.0 mg/mL G-418).

Receptor was expressed and purified following the previously described protocol (*Niebauer and Robinson, 2006*). In brief, from freshly streaked YPD plates (1% yeast extract, 2% peptone, 2% dextrose), single colonies were grown in 5 mL YPD cultures overnight at 30°C. From these 5 mL cultures, 50 mL cultures were grown with a starting OD of 0.5 overnight at 30°C. To induce expression, yeast cells from these 50 mL cultures were centrifuged at 3000 × *g* to remove YPD before resuspended in YPG medium (1% yeast, 2% peptone, 2% D-galactose) at a starting OD of 0.5. The receptor was expressed for 24 hr overnight at 30°C with 250 rpm shaking. Cells were pelleted by centrifugation at 3000 × *g*, washed in sterile PBS buffer, and pelleted again before storage at –80°C until purification.

Mechanical bead lysis of cells was done, per 250 mL of cell culture, by performing 12 pulses of 60 s intense vortexing (with at least 60 s of rest in between pulses) in 10 mL 0.5 mm zirconia silica beads (BioSpec, Bartlesville, OK, USA; #11079105z), 25 mL of lysis buffer (50 mM sodium phosphate, 300 mM sodium chloride, 10% [v/v] glycerol, pH = 8.0, 2% [w/v] n-dodecyl-β-D-maltopyranoside [DDM; Anatrace, Maumee, OH, USA; #D310], 1% [w/v] 3-[(3-cholamidopropyl)dimethylammonio]−1-propanesulfonate [CHAPS; Anatrace; #C216], and 0.2% [w/v] cholesteryl hemisuccinate [CHS; Anatrace; #CH210] and an appropriate amount of 100× Pierce Halt EDTA-free protease inhibitor [Pierce, Rockford, IL, USA; #78439]). Beads were separated using a Kontex column. Unlysed cells were removed by centrifugation at 3220 × *g* for 10 min. Receptor was let solubilized on rotary mixer for 3 hr before cell debris was removed by centrifugation at 10,000 × *g* for 30 min. Solubilized protein was incubated with Ni-NTA resin (Pierce; #88221) overnight. Protein-resin mixture was then washed extensively in purification buffer (50 mM sodium phosphate, 300 mM sodium chloride, 10% [v/v] glycerol, 0.1% [w/v] DDM, 0.1% [w/v] CHAPS and 0.02% [w/v] CHS, pH = 8.0) containing low imidazole concentrations (20–50 mM). A$_{2A}$R was eluted into purification buffer containing 500 mM imidazole. Prior to further chromatographic purification, imidazole was removed using a PD-10 desalting column (GE Healthcare, Pittsburgh, PA, USA; #17085101).

Ligand affinity resin was prepared as previously described for purification of active A$_{2A}$R (*O'Malley et al., 2007*; *Weiß and Grisshammer, 2002*). In brief, 8 mL of isopropanol-washed Affigel 10 resin (BioRad, Hercules, CA, USA; #1536099) was mixed gently in an Erlenmeyer flask for 20 hr at room temperature with 48 mL of DMSO containing 24 mg of xanthine amine congener (XAC, high-affinity A$_{2A}$R antagonist, K$_D$ = 32 nM; Sigma, St. Louis, MO, USA; #X103). The absorbance at 310 nm of the XAC-DMSO solution before and after the coupling reaction was measured in 10 mM HCl and compared to a standard curve. The amount of resin bound to ligand was estimated to be 5.6 µM. The coupling reaction was quenched by washing the resin with DMSO, then with Tris-HCl 50 mM (pH = 7.4), then with 20% (v/v) ethanol. The resin was packed into a Tricorn 10/50 column (GE Healthcare) under pressure via a BioRad Duoflow FPLC (BioRad).

For purification of active A$_{2A}$R, the column was equilibrated with 4 CV of purification buffer. The IMAC-purified A$_{2A}$R was desalted and diluted to 5.5 mL before applied to a 5 mL sample loop on the BioRad Duoflow FPLC, from which the sample was loaded onto the column at a rate of 0.1 mL/min. Inactive A$_{2A}$R was washed from the column by flowing 10 mL of purification buffer at 0.2 mL/min, followed by 16 mL at 0.4 mL/min. Active A$_{2A}$R was eluted from the column by flowing purification buffer containing 20 mM theophylline (low-affinity A$_{2A}$R antagonist, K$_D$ = 1.6 µM; Sigma; #T1633). Western blot analysis was performed to determine 4 mL fractions with active A$_{2A}$R collected with a BioFrac fraction collector (BioRad), which were then concentrated through a 30 kDa MWCO centrifugal filter (Millipore, Billerica, MA, USA; #UFC803096) and desalted to remove excess theophylline. For the experiments where the salt concentrations were varied, the buffer exchange was done also by this last desalting step.

## Size-exclusion chromatography

To separate oligomeric species of active $A_{2A}R$, a prepacked Tricorn Superdex 200 10/300 GL column (GE Healthcare; #17-5175-01) connected to a BioRad Duoflow FPLC was equilibrated with 60 mL of running buffer (150 mM sodium chloride except for the ionic strength experiments where NaCl concentration is adjusted to achieve the desired ionic strengths, 50 mM sodium phosphate, 10% [v/v] glycerol, 0.1% [w/v] DDM, 0.1% [w/v] CHAPS, 0.02% [w/v] CHS, pH = 8.0) at a flow rate of 0.2 mL/min. 0.5 mL fractions were collected with a BioFrac fraction collector in 30 mL of running buffer at the same flow rate. The subsequent SEC analysis performed on the SEC-separated oligomeric populations also followed this protocol.

## SEC peak analysis

SEC chromatograms were analyzed using OriginLab using the nonlinear curve fit (Gaussian) function. The area under the curve and the peak width were manually defined in cases where the SNR of the SEC trace were too low. The $R^2$ values reached > 0.96 for most cases. The population of each oligomeric species was expressed as the integral of each Gaussian this curve fit of the SEC signal. The HMW oligomer peak in some cases could not be fitted with one curve and thus was fitted with two curves instead. The reported standard errors were calculated from the variance of the fit and did not correspond to experimental errors. The results are detailed in *Figure 1—figure supplement 2* and *Supplementary file 1*.

## SDS-PAGE and western blotting

10% SDS-PAGE gels were hand-casted in BioRad Criterion empty cassettes (BioRad; #3459902, 3459903). Lysate controls were prepared by lysis of 5 OD cell pellets with 35 μL of YPER (Fisher Scientific, Waltham, MA, USA; #8990) at RT for 20 min, incubation with 2× Laemmli buffer (4% [w/v] SDS, 16% [v/v] glycerol, 0.02% [w/v] bromophenol blue, 167 M Tris, pH 6.8) at 37°C for 1 hr, and centrifugation at 3000 × *g* for 1 min to pellet cell debris. Protein samples were prepared by incubation with 2× Laemmli buffer at 37°C for 30 min. For all samples, 14 μL (for 26-well gel) or 20 μL (for 18-well gel) was loaded per lane, except for 7 μL of Magic Mark XP Western protein ladder (Thermo Scientific, Waltham, MA, USA; #LC5602) as a standard. Electrophoresis was carried out at 120 V for 100 min. Proteins were transferred to 0.2 μm nitrocellulose membranes (BioRad; #170-4159) via electroblotting using a BioRad Transblot Turbo, mixed MW protocol. Membranes were blocked in Tris-buffered saline with Tween (TBST; 150 mM sodium chloride, 15.2 mM Tris-HCl, 4.6 mM Tris base, pH = 7.4, 0.1% [v/v] Tween 20 [BioRad; #1706531]) containing 5% (w/v) dry milk, then probed with anti-$A_{2A}R$ antibody, clone 7F6-G5-A2, mouse monoclonal (Millipore, Burlington, MA, USA; #05-717) at 1:500 in TBST with 0.5% (w/v) dry milk. Probing with secondary antibody was done with a fluorescent anti-mouse IgG H&L DyLight 550 antibody (Abcam, Cambridge, MA, USA; #ab96880) at 1:600 in TBST containing 0.5% (w/v) milk.

Western blot was analyzed with Image Lab 6.1 software (BioRad), with built-in tool to define each sample lane and to generate an intensity profile. Peaks were manually selected and integrated with the measure tool to determine the amount of protein present.

## CGMD simulations

Initial configuration of $A_{2A}R$ was based on the crystal structure of the receptor in the active state (PDB 5G53). Since this structure does not include the entire C-terminus, we resorted to using homology modeling software (i.e., MODELLER 9.23) (*Eswar et al., 2006*) to predict the structures of the C-terminus. After removing all non-receptor components, the first segment of the C-terminus consisting of residues 291–314 was modeled as a helical segment parallel to the cytoplasmic membrane surface while the rest of the C-terminus was modeled as intrinsically disordered. MODELLER is much more accurate in structural predictions for segments less than 20 residues. This limitation necessitated that we run an equilibrium MD simulation for 2 μs to obtain a well-equilibrated structure that possesses a more viable starting conformation. To validate our models of all potential variants of $A_{2A}R$, we calculated the RMSD and RMSF for each respective system. Default protonation states of ionizable residues were used. The resulting structure was converted to MARTINI CG topology using the martinize.py script (*de Jong et al., 2013*). The ELNeDyn elastic network (*Periole et al., 2009*) was used to constrain protein secondary and tertiary structures with a force constant of 500 kJ/mol/

$nm^2$ and a cutoff of 1.5 nm. To optimize loop refinement of the model, a single copy was embedded in a 1-palmitoyl-2-oleoyl-sn-glycero-3-phosphocholine (POPC) bilayer using the insane.py script, solvated with MARTINI polarizable water, neutralized with 0.15 M NaCl, and a short MD (1.5 µs) run to equilibrate the loop regions. Subsequently, two monomers of the equilibrated $A_{2A}R$ were randomly rotated and placed at the center of a 13 nm × 13 nm × 11 nm (xyz) box, 3.5 nm apart, with their principal transmembrane axis aligned parallel to the z axis. The proteins were then embedded in a POPC bilayer using the insane.py script. Sodium and chloride ions were added to neutralize the system and obtain a concentration of 0.15 M NaCl. Total system size was typically in the range of 34,000 CG particles, with a 280:1 lipid:protein ratio. Ten independent copies were generated for each $A_{2A}R$ truncated variant. v2.2 of the MARTINI CG force field (*Monticelli et al., 2008*) was used for the protein and water, and v2.0 was used for POPC. All CG simulations were carried out in GRO-MACS 2016 (*Abraham et al., 2015*) in the NPT ensemble (P = 1 atm, T = 310 K). The Bussi velocity rescaling thermostat was used for temperature control with a coupling constant of $\tau_t$ = 1.0 ps (*Bussi et al., 2007*), while the Parrinello–Rahman barostat (*Martonák et al., 2003*) was used to control the pressure semi-isotropically with a coupling constant of $\tau_t$ = 12.0 ps and compressibility of $3 \times 10^{-4}$ $bar^{-1}$. Reaction field electrostatics was used with Coulomb cutoff of 1.1 nm. Non-bonded Lennard–Jones interactions were treated with a cutoff of 1.1 nm. All simulations were run with a 15 fs time step, updating neighbor lists every 10 steps. Cubic periodic boundary conditions along the x, y, and z axes were used. Each simulation was run for 8 µs.

## Atomistic MD simulations

Three snapshots of symmetric dimers of $A_{2A}R$ for each respective truncated variant were randomly selected from the CG simulations as starting structures for backmapping. CG systems were converted to atomistic resolution using the backward.py script (*Wassenaar et al., 2014*). All simulations were run in Gromacs2019 in the *NPT* ensemble (P = 1 bar, T = 310 K) with all bonds restrained using the LINCS method (*Hess et al., 1997*). The Parrinello–Rahman barostat was used to control the pressure semi-isotropically with a coupling constant of $\tau_t$ = 1.0 ps and a compressibility of $4.5 \times 10^{-5}$ $bar^{-1}$, while the Bussi velocity rescaling thermostat was used for temperature control with a coupling constant of $\tau_t$ = 0.1 ps. Proteins, lipids, and solvents were separately coupled to the thermostat. The CHARMM36 and TIP3P force fields (*Best et al., 2012*; *Jorgensen et al., 1983*) were used to model all molecular interactions. Periodic boundary conditions were set in the x, y, and z directions. Particle mesh Ewald (PME) electrostatics was used with a cutoff of 1.0 nm. A 2-fs time step was used for all atomistic runs, and each simulation was run for 50 ns.

## Analysis of computational results

All trajectories were postprocessed using gromacs tools and in-house scripts. We ran a clustering analysis of all dimer frames from the CG simulations using Daura et al.'s clustering algorithm (*Daura et al., 1999*) implemented in GROMACS, with an RMSD cutoff of 1.5 Å. An interface was considered dimeric if the minimum center of mass distance between the protomers was less than 5 Å. This method uses an RMSD cutoff to group all conformations with the largest number of neighbors into a cluster and eliminates these from the pool, then repeats the process until the pool is empty. We focused our analysis on the most populated cluster from each truncated variant. Electrostatic interactions in the dimer were calculated from CG systems with LOOS (*Romo and Grossfield, 2009*) using a distance cutoff of 5.0 Å. Transmembrane helical tilt angles were also calculated in LOOS from CG simulations. Hydrogen bonds were calculated from AA simulations using the hydrogen bonds plugin in VMD (*Humphrey et al., 1996*), with a distance cutoff of 3.5 Å and an angle cutoff of 20°. Only C-terminal residues were included in hydrogen bond analysis. PyMOL (The PyMOL Molecular Graphics System, version 2.0, *Schrödinger, LLC, 2020*) was used for molecular visualizations.

## Assessing $A_{2A}R$ oligomerization with increasing ionic strength

$Na_2HPO_4$ and $NaH_2PO_4$ in the buffer make up an ionic strength of 0.15 M, to which NaCl was added to increase the ionic strength to 0.45 M and furthermore to 0.95 M. The $A_{2A}R$ variants were purified at 0.45 M ionic strength and then exchanged into buffers of different ionic strengths using a PD-10 desalting column prior to subjecting the samples to SEC. The buffer composition is detailed below.

| Buffers | Components | Concentration (mM) | Ionic strength (mM) |
|---|---|---|---|
| 0.15 M ionic strength | NaCl | 0 | 0 |
| | NaH$_2$PO$_4$ | 4 | 4 |
| | Na$_2$HPO$_4$ | 49 | 146 |
| 0.45 M ionic strength | NaCl | 300 | 300 |
| | NaH$_2$PO$_4$ | 4 | 4 |
| | Na$_2$HPO$_4$ | 49 | 146 |
| 0.95 M ionic strength | NaCl | 800 | 800 |
| | NaH$_2$PO$_4$ | 4 | 4 |
| | Na$_2$HPO$_4$ | 49 | 146 |

### Isolated C-terminus purification

*Escherichia coli* BL21 (DE3) cells (Sigma; #CMC0014) were transfected with pET28a DNA plasmids containing the desired A$_{2A}$R sequence with a 6x His tag attached for purification. Cells from glycerol stock were grown in 10 mL luria broth (LB, Sigma Aldrich, L3022) overnight at 37°C and then used to inoculate 1 L of fresh LB and 10 μg/mL kanamycin (Fisher Scientific, BP906). Growth of cells was performed at 37°C, 200 rpm until optical density at λ = 600 nm reached 0.6–0.8. Expression was induced by incubation with 1 mM isopropyl-β-D-thiogalactoside (Fisher Bioreagents, BP175510) for 3 hr.

Cells were harvested with centrifugation at 5000 rpm for 30 min. Harvested cells were resuspended in 25 mL Tris-HCl, pH = 7.4, 100 mM NaCl, 0.5 mM DTT, 0.1 mM EDTA with 1 Pierce protease inhibitor tablet (Thermo Scientific, A32965), 1 mM PMSF, 2 mg/mL lysozyme, 20 μg/mL DNase (Sigma, DN25) and 10 mM MgCl$_2$, and incubated on ice for 30 min. Samples were then incubated at 30°C for 20 min, then flash frozen and thawed three times in LN$_2$. Samples were then centrifuged at 10,000 rpm for 10 min to remove cell debris. 1 mM PMSF was added again and the resulting supernatant was incubated while rotating for at least 4 hr with Ni-NTA resin. The resin was loaded to a column and washed with 25 mL 20 mM sodium phosphate, pH = 7.0, 1 M NaCl, 20 mM imidazole, 0.5 mM DTT, 100 μM EDTA. Purified protein was eluted with 15 mL of 20 mM sodium phosphate, pH = 7.0, 0.5 mM DTT, 100 mM NaCl, 300 mM imidazole. The protein was concentrated to a volume of 2.5 mL and was buffer exchanged into 20 mM ammonium acetate buffer, pH = 7.4, 100 mM NaCl using a GE PD-10 desalting column. Purity of sample was confirmed with SDS-PAGE and western blot.

### Aggregation assay to assess A$_{2A}$R C-terminus assembly

Absorbance was measured at 450 nm using a Shimadzu UV-1601 spectrophotometer with 120 μL sample size. Prior to reading, samples were incubated at 40°C for 5 min. Samples were vigorously pipetted to homogenize any precipitate before absorbance was measured. Protein concentration was 50 μM in a 20 mM ammonium acetate buffer (pH = 7.4).

### Differential scanning fluorimetry (DSF)

DSF was conducted with a BioRad CFX90 real-time PCR machine. A starting temperature of 20°C was increased at a rate of 0.5°C per 30 s to a final temperature of 85°C. All samples contained 40 μL of 40 μM A$_{2A}$R C-terminus, 9x SYPRO orange (ThermoFisher S6650), 200 mM NaCl, and 20 mM MES. Fluorescence was detected in real time at 570 nm. All samples were conducted in triplicate.

### Hydrophobicity and charge profile of C-terminus

The hydrophobicity profile reported in *Figure 6—figure supplement 1* was determined with ProtScale using method described by *Kyte and Doolittle, 1982*, window size of 3.

## Acknowledgements

This material is based upon work supported by (1) the National Institute of General Medical Sciences of the National Institutes of Health under Award Number R35GM136411, (2) the National Institute of Mental Health of the National Institutes of Health under Small Business Innovation Research Award Number 1R43MH119906-01, and (3) the National Science Foundation under Award Number MCB-1714888 (ES and BM). The content is solely the responsibility of the authors and does not necessarily represent the official views of the National Institutes of Health. Many of the experiments were completed with the assistance from Rohan Katpally. The pITy expression vector and *S. cerevisiae* BJ5464 strain were generously provided by Prof. Anne Robinson's lab at Carnegie Mellon University. The X7 polymerase was a gift from Dr. Morten Nørholm, Novo Nordisk Foundation Center for Biosustainability, Technical University of Denmark. Computational time was provided through WVU Research Computing and XSEDE allocation no. TG-MCB130040.

## Additional information

### Funding

| Funder | Grant reference number | Author |
| --- | --- | --- |
| National Institute of General Medical Sciences | R35GM136411 | Khanh Dinh Quoc Nguyen<br>Michael Vigers<br>Susanna Seppälä<br>Nicole Star Schonenbach<br>Michelle Ann O'Malley<br>Songi Han |
| National Institute of Mental Health | 1R43MH119906-01 | Khanh Dinh Quoc Nguyen<br>Jennifer Paige Hoover<br>Michelle Ann O'Malley<br>Songi Han |
| National Science Foundation | MCB-1714888 | Eric Sefah<br>Blake Mertz |

The funders had no role in study design, data collection and interpretation, or the decision to submit the work for publication.

### Author contributions

Khanh Dinh Quoc Nguyen, Conceptualization, Data curation, Formal analysis, Validation, Investigation, Visualization, Methodology, Writing - original draft, Writing - review and editing; Michael Vigers, Conceptualization, Data curation, Formal analysis, Validation, Investigation, Writing - review and editing; Eric Sefah, Conceptualization, Data curation, Software, Formal analysis, Validation, Investigation, Visualization, Methodology, Writing - review and editing; Susanna Seppälä, Conceptualization, Investigation, Writing - review and editing; Jennifer Paige Hoover, Data curation, Investigation; Nicole Star Schonenbach, Conceptualization, Data curation, Formal analysis, Investigation, Writing - review and editing; Blake Mertz, Conceptualization, Software, Supervision, Funding acquisition, Validation, Writing - review and editing; Michelle Ann O'Malley, Conceptualization, Supervision, Funding acquisition, Writing - review and editing; Songi Han, Conceptualization, Supervision, Funding acquisition, Writing - original draft, Writing - review and editing

### Author ORCIDs

Khanh Dinh Quoc Nguyen https://orcid.org/0000-0002-9367-499X
Songi Han https://orcid.org/0000-0001-6489-6246

### Decision letter and Author response

Decision letter https://doi.org/10.7554/eLife.66662.sa1
Author response https://doi.org/10.7554/eLife.66662.sa2

## Additional files

### Supplementary files

• Supplementary file 1. Results from curve fitting using OriginLab and calculations of the high-molecular-weight (HMW) oligomer and dimer levels for all $A_{2A}R$ variants used in the main text of this study. The variants are grouped by the order they appear and numbered corresponding to *Figure 1—figure supplement 2*. The levels of dimer and HMW oligomer are expressed relative to the monomeric population in arbitrary units as monomer-equivalent concentration ratios. The errors are calculated from the variance of the fit, not experimental variations, and are within 95% confidence interval. Only the WT replicates are represented with standard deviation as experimental variations (last row; n = 5; mean ± SD).

• Transparent reporting form

### Data availability

All data generated or analysed during this study are included in the manuscript and supporting files.

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
