## [Decision Letter]

**Acceptance summary:**

This is a wonderful study that advances our understanding of GPCR oligomerization and provides new physical insights into GPCR-mediated cellular signaling.

**Decision letter after peer review:**

Thank you for submitting your article "Oligomerization of the Human Adenosine A2A Receptor Is Driven by the Intrinsically Disordered C-Terminus" for consideration by *eLife*. Your article has been reviewed by 3 peer reviewers, including Heedeok Hong as the Reviewing Editor and Reviewer #1, and the evaluation has been overseen by Olga Boudker as the Senior Editor. The following individual involved in review of your submission has agreed to reveal their identity: Antonella Di Pizio (Reviewer #3).

Essential revisions:

1. On the rigor and validity of size-exclusion chromatography (SEC)

(1a) Justifying the peak assignment in the SEC data as monomer, dimer and HMW oligomers (Figure 1 and Figure S1):

While the UV signals on the SEC profiles suggest the existence of well-resolved peaks of oligomers, dimers and monomers, the SDS-PAGE and Western blotting results that were carried out in a denaturing environment (SDS) predominantly display monomers in the "dimer" and "HMW" fractions. An alternative method may be needed to verify the assignment (e.g., crosslinking, an assignment based on the standard curve-i.e., mobility vs MW standard, analytical ultracentrifuge, native gel, etc.).

(1b) Ensuring that oligomer distributions are thermodynamic products:

The clarification of this point seems necessary to support the conclusion that multiple types of molecular forces serve as "driving forces" in oligomerization. Probably, it would be helpful to rerun SEC for the fractions of each major peak (possibly C394X mutants) and investigate the dependence of oligomer distribution on protein and detergent concentrations, the presence of deca-His tag and the length of storage. It would also be important to confirm that "UV in arbitrary units" scales with the protein concentration in the fractions.

(1c) Strengthening the rigor of statistical analysis:

Reported experimental uncertainties of content of oligomerized receptor fractions are solely based on reproducibility of fits of a single elution profile. It may underestimate experimental error limits. How reproducible are results when chromatographic experiments are repeated using the same protein stock?

2. Verifying the influence of mutation and C-terminal truncation on ligand binding capability.

3. Providing further details and additional analysis of MD simulation.

(3a) More detailed descriptions for initial modeling and its evaluation procedures (please see Reviewer 3's comments). For example, which was(were) template(s) used for the modelling of the initial state and how was the reliability of the model evaluated?

(3b) Please, address reviewers' concerns about how the system equilibration was ensured (simulation sufficiently long or repeated sufficiently with a variation of initial conditions to yield results that are not biased by initial conditions) and report time-dependent RMSDs that can provide the information on the equilibration and dynamics of the system.

(3c) Additional analysis will help further strengthening the conclusion: (i) statistical analysis of properties of interaction sites between neighbored molecules; (ii) the role of protein segments other than the C-terminus for oligomerization; (iii) statistical analysis of intra- and intermolecular "nonpolar" contacts to support authors' claim that the hydrophobic interaction is one of the key driving forces in oligomerization.

4. Clarifying the putative disulfide bridge involving C394.

It would be useful to provide a list of Cys residues that potentially can interact with Cys394 and discuss the validity of authors' claim on the formation of putative disulfide bridge.

5. Addressing additional major scientific concerns from reviewers:

(5a) Is the result obtained in micelles relevant in the lipid bilayer or cellular context?

(5b) What is the basis of claiming the "cooperativity" in the C-terminal domain interactions?

6. Please, address reviewers' major concerns that have been brought up (see below) but not listed above.

*Reviewer #1 (Recommendations for the authors):*

I have several suggestions that may help.

1. How relevant is the peak assignment in the SEC data as monomer, dimer and HMW oligomers (Figure 1)?

Although the assignment is supported by SDS-PAGE and Western blotting (Figure S1), SDS provides a denaturing environment. As seen in the data (Figure S1), the proteins in the "dimer" and "HMW" fractions on SEC dominantly migrate as monomers on SDS-PAGE, which indicates that SDS destabilizes oligomers or, if not, each peak contains a significant portion of monomer. An alternative method may be needed to verify the assignment (for example, crosslinking, an assignment based on the standard curve- mobility vs MW standards, analytical ultracentrifuge or native gel).

2. Ensuring that the oligomer distributions are thermodynamic products.

Probably, it might be helpful to rerun SEC with the fractions of each major peak (possibly C394X mutants) and see if the redistribution of oligomeric states occurs.

3. On the contribution of the hydrophobic interactions to oligomerization.

(3a) Since it has been suggested that the hydrophobic effect is one of the key driving forces for oligomerization, it would be informative to (3a-i) show the fraction of nonpolar residues in the C-term tail and (3a-ii) analyze the number of nonpolar contacts during CGMD simulations

(3b) What are the RMSDs of the IDRs during simulation? This analysis will be highly informative with regards to the equilibration of the system, the dynamics of the IDR, and the effect of truncation on the dynamics.

*Reviewer #2 (Recommendations for the authors):*

I am concerned about inconsistencies between UV absorbance and Western Blot analysis of eluted fractions in Figure S1B. While the UV signal suggests existence of well-resolved peaks of oligomers, dimers and monomers, the Western blots show high concentration of monomers underneath the dimer peak. What is the cause for this discrepancy? What are the protein concentration applied to the column and concentrations in the eluted fractions? Does aggregation behavior depend on protein concentration, detergent concentration, temperature, length of storage? Did the protein denature partially on the column? Did the authors repeat experiments on concentrated eluted fractions? Could it be that baseline correction of UV absorbance obscured a broad peak of monomer elution?

The use of the term "UV in arbitrary units" when reporting ratios of protein in oligo-, di- and monomers is ambiguous. Does protein concentration in the fractions scale with integral intensity of UV absorbance traces? If yes, the ratios would faithfully report relative differences in protein content of those fractions.

Reported experimental uncertainties of content of oligomerized receptor fractions are solely based on reproducibility of fits of a single elution profile. It yields experimental error limits that are rather low. How reproducible are results when chromatographic experiments are repeated using the same protein stock?

The authors report evidence for disulfide bond formation between neighbored molecules at C394. The level of disulfide bond formation is known to depend on cofactors including protein concentration, oxygen exposure, pH, the presence of oxidizing or reducing agents, temperature, time, etc. Were those variables controlled?

Does the deca-His tag influence oligomerization of the protein?

Does truncation of the receptor influence its function? Did the authors observe differences in ligand binding affinity and G protein activation rates for truncated/mutated receptor? Does protein truncation influence expression yield? Does truncation influence thermal stability of the expressed protein? Are eluted protein fractions obtained by size exclusion chromatography ligand binding- and G protein activation competent?

Experimental results are accompanied by an impressive set of molecular simulations. Were simulations sufficiently long or repeated sufficiently often with variation of initial conditions to yield results that are not biased by initial conditions? Would it be possible to conduct a statistical analysis of properties of interaction sites between neighbored molecules? What is the role of protein segments other than the C-terminus for aggregation?

*Reviewer #3 (Recommendations for the authors):*

1. The putative disulfide bridge involving C394 should be further investigated.

– In light of the suggested C-terminus/C-terminus interaction in absence of the TM domain, the Cys partner might be in the C-terminus. It would be useful to provide a list of Cys residue that potentially can interact with Cys394 and experimentally validate these hypotheses

– The discussion in lines 338-391 should be extended accordingly: 'A previous study showed that residue C394 in A2AR dimer is available for nitroxide spin labelling(Schonenbach et al. 2016), suggesting that some of these disulfide bonds may be between 390 residue C394 and another cysteine in the hydrophobic core of A2AR that do not form intramolecular disulfide bonds(De Filippo et al. 2016; Naranjo et al. 2015; O'Malley et al. 2010)'

– Moreover, in some parts of the manuscript the putative disulfide bridge is ignored, es: Lines 86-87: 'a model GPCR that could engage in diverse non-covalent interactions, such as electrostatic interactions, hydrogen bonds, or hydrophobic interactions. These non covalent interactions are readily tunable by external factors', Lines 291-293: 'The variable 13 291 nature of A2AR oligomeric interfaces suggests that the main driving forces must be non-covalent interactions, such as electrostatic interactions and hydrogen bonds as identified by the above MD simulations'

2. MD simulations

The C-terminus is not present in any of the A2AR crystal structures and is very long (Lines 104-105: A striking example is A2AR, a model GPCR with a particularly long, 122-residue, C-terminus that is truncated in all published structural biology studies).

The C-terminus is therefore modelled, however, the only reference to the C-terminus modelling I could find is in Lines 594-595: 'missing residues added using MODELLER 9.23(Eswar et al. 2006)'. Which template(s) was(were) used for the modelling and which is the sequence similarity? The detailed modelling procedure and the computational evaluation should be provided.

Results of the MD are highly dependent of the input model. Moreover, the information about the disulfide bridge is not incorporated in the models but this is an important structural feature to be considered.

Also, the conclusions about the role of the ERR motif are based on the modelling, but we do not have information to judge the modelling.

Lines 409-410: 'This observation is supported by our experimental results showing that substituting this charged cluster with alanines reduces the total A2AR oligomer levels' – the experimental results suggest the involvement of these residues on the oligomerization process, but do not say a lot about the molecular mechanisms – localizing these residues far from the interacting surface and the intramolecular interactions are hypotheses based on the modelling.

3. The impact of findings is weakly stated, some related sentences in the paper are very general:

– Line 38 in the abstract: 'offering important guidance for structure-function studies of A2AR and other GPCRs'

– Lines 55-56: 'it is crucial to identify the driving factors that govern the oligomerization of GPCRs, such that the properties of GPCR oligomers can be understood'

– Lines 473-475: 'In that context, this study offers valuable insights and approaches to tune the oligomerization of A2AR and potentially of other GPCRs using its intrinsically disordered C-terminus'

4. I suggest labeling TM residues with BW numbering, so it will be easier to distinguish between TM residues and C-terminus residues in the figures and the text.

5. Title: 'homo-oligomerization' should replace 'oligomerization'

---

## [Author Response]

Reviewer #1 (Recommendations for the authors):I have several suggestions that may help.1. How relevant is the peak assignment in the SEC data as monomer, dimer and HMW oligomers (Figure 1)?Although the assignment is supported by SDS-PAGE and Western blotting (Figure S1), SDS provides a denaturing environment. As seen in the data (Figure S1), the proteins in the "dimer" and "HMW" fractions on SEC dominantly migrate as monomers on SDS-PAGE, which indicates that SDS destabilizes oligomers or, if not, each peak contains a significant portion of monomer. An alternative method may be needed to verify the assignment (for example, crosslinking, an assignment based on the standard curve- mobility vs MW standards, analytical ultracentrifuge or native gel).

We are aware of this issue, which is why we did not use SDS-PAGE and Western blotting results as the primary method to assign the SEC peaks. Instead, we used these methods to verify that the protein is pure and indeed corresponds to A_2A_R.

Rigorous experiments have been done to justify the peak assignment in the SEC data. The surfactant used in our study can bind to solubilized receptors and alter both the apparent molecular weight and the hydrodynamic radius of the receptors, thus compromising our ability to estimate the size of the eluting species by comparing to SEC standards. For this reason, we used multiangle light scattering (MALS) coupled with SEC to estimate the molecular weights of the SEC-separated A_2A_R species. The analyses of A_2A_R-WT showed that the approximate molecular weights of the monomer, dimer, and HMW oligomer are 49.5, 109.2, and 332.3 kDa. These data compared well with the expected molecular weights of A_2A_R monomer (46.8 kDa), dimer (93.6 kDa), and heptamer (327.6 kDa). The results were published in one of our previous studies (Schonenbach et al., FEBS Lett 2016, 590, 3295–3306).

2. Ensuring that the oligomer distributions are thermodynamic products.Probably, it might be helpful to rerun SEC with the fractions of each major peak (possibly C394X mutants) and see if the redistribution of oligomeric states occurs.

We did perform new experiments to test the stability and reversibility of the A_2A_R monomer and dimer/oligomer population, of both the A_2A_R-WT and A_2A_R-Q372ΔC variants (Figure 5—figure supplement 1A). We find that the SEC-separated monomers repopulate measurably into dimer/oligomer, with the total oligomer level after redistribution comparable with that of the initial samples for both A_2A_RWT (initial: 2.87; redistributed: 1.60) and A_2A_R-Q372ΔC (initial: 1.49; redistributed: 1.40) (Figure 5—figure supplement 1A). This observation indicates that A_2A_R oligomer is a thermodynamic product with a lower free energy compared with that of the monomer. This is consistent with the results we have shown in the manuscript that the oligomer levels of A_2A_R-WT are consistent (1.34–2.87; Table S1) and that A_2A_R oligomerization can be modulated with ionic strengths via depletion interactions (Figure 5).

Interestingly, the SEC-separated dimer/oligomer populations do not repopulate to form monomers (Figure 5—figure supplement 1A). This observation is consistent with a published study of ours on A_2A_R dimers (Schonenbach et al., FEBS Lett 2016, 590, 3295–3306). This indicates that once the oligomers are formed, some are kinetically trapped and thus cannot redistribute into monomers. We believe that it is disulfide linkages that kinetically stabilize A_2A_R oligomers, as demonstrated by their redistribution into monomers only in the presence of a reducing agent (Figure 2B).

Taken together, we suggest that A_2A_R oligomerization is a thermodynamic process (Figure 5—figure supplement 1B), with the monomer overcoming the activation energy (E_A_) by depletion interactions to repopulate into dimer/oligomer with a slightly lower free energy. Once formed, the redistributed dimer/oligomer populations can be kinetically stabilized by disulfide linkages. The results are summarized in the figure now included in the manuscript as Figure 5—figure supplement 1. We will also include this argument in the Discussion section.

3. On the contribution of the hydrophobic interactions to oligomerization.(3a) Since it has been suggested that the hydrophobic effect is one of the key driving forces for oligomerization, it would be informative to (3a-i) show the fraction of nonpolar residues in the C-term tail and (3a-ii) analyze the number of nonpolar contacts during CGMD simulations

(i) Thank you for your suggestion. We have included a new figure to show this, and we also refer to this figure in our manuscript now as Figure 6—figure supplement 1B.

(ii) We analyzed the number of nonpolar contacts in our CGMD simulations and found that there is a general correlation between the length of the C-terminus and the number of contacts between nonpolar residues. The results are now included in the manuscript as Figure 6—figure supplement 1C. Since these interactions are very weak in nature and further diminished due to the coarse-graining resolution, we need to further investigate this issue via atomistic MD simulations (which is outside the scope of the current study).

(3b) What are the RMSDs of the IDRs during simulation? This analysis will be highly informative with regards to the equilibration of the system, the dynamics of the IDR, and the effect of truncation on the dynamics.

We agree with the reviewer’s comment and have conducted this analysis (see our detailed answer above).

Reviewer #2 (Recommendations for the authors):I am concerned about inconsistencies between UV absorbance and Western Blot analysis of eluted fractions in Figure S1B. While the UV signal suggests existence of well-resolved peaks of oligomers, dimers and monomers, the Western blots show high concentration of monomers underneath the dimer peak. What is the cause for this discrepancy? What is the protein concentration applied to the column and concentrations in the eluted fractions? Does aggregation behavior depend on protein concentration, detergent concentration, temperature, length of storage? Did the protein denature partially on the column? Did the authors repeat experiments on concentrated eluted fractions? Could it be that baseline correction of UV absorbance obscured a broad peak of monomer elution?

Thank you for your questions. We will answer each one in order:

– The discrepancy is caused by SDS partially destabilizing the oligomeric species. Therefore, the SDS-PAGE and Western blotting results are only used to demonstrate the purity and the identity of the purified A_2A_R, and not to justify peak assignment. Rigorous experiments have been done to justify the peak assignment in the SEC data. We used multiangle light scattering (MALS) coupled with SEC to estimate the molecular weights of the SEC-separated A_2A_R species. The analyses of A_2A_R-WT showed that the approximate molecular weights of the monomer, dimer, and HMW oligomer are 49.5, 109.2, and 332.3 kDa. These data compared well with the expected molecular weights of A_2A_R monomer (46.8 kDa), dimer (93.6 kDa), and heptamer (327.6 kDa). The results were published in one of our previous studies (Schonenbach et al., FEBS Lett 2016, 590, 3295–3306). We now have these experiment referred to in line 153–155 in the manuscript for clarification.

– The concentration before SEC is about 5 mg/mL, while after SEC it is diluted about 20-fold.

– We have not done any experiment that shows changes in the oligomerization pattern with protein concentration, detergent concentration, temperature, or length of storage. The protein concentration is varied between experiments, but there has not been any correlation with dimer/oligomer levels. Detergent concentration and temperature are always kept the same. If the protein is stored for too long (~1 week), the C-terminus is cleaved off, certainly excluding oligomerization; hence, we only use freshly prepared proteins.

– We have not seen any evidence of denaturation.

– Yes, we have run SEC experiments on SEC-separated dimer/oligomer vs. monomer fractions of A_2A_R-WT and Q372ΔC (Figure 5—figure supplement 1). We found that the dimer/oligomer population from SEC elution remained dimer/oligomer, with little to no redistribution into other oligomeric states. This implies that dimer formation is a protein-intrinsic property of A_2A_R, not a property imposed by the solution environment. Meanwhile, the monomer population did repopulate into oligomers. Taken together, these results imply that the formed dimers are thermodynamically highly stable, state, while the formation of A_2A_R dimer/oligomer is an activated process that requires the lowering of an energy barrier to proceed.

– There should not be any broad trailing monomer peak after that. We have run Western blots of fractions of fractions eluted after the monomer fractions and detected no protein.

The use of the term "UV in arbitrary units" when reporting ratios of protein in oligo-, di- and monomers is ambiguous. Does protein concentration in the fractions scale with integral intensity of UV absorbance traces? If yes, the ratios would faithfully report relative differences in protein content of those fractions.

Thank you for your question. Yes, the protein concentration does scale with integral intensity of UV absorbance traces.

Reported experimental uncertainties of content of oligomerized receptor fractions are solely based on reproducibility of fits of a single elution profile. It yields experimental error limits that are rather low. How reproducible are results when chromatographic experiments are repeated using the same protein stock?

We are aware that the reported uncertainties are low. Therefore, we repeatedly emphasized in the manuscript that the uncertainties are the results of the curve-fitting process, and not experimental errors. Limited time and resources during the pandemic (UCSB research operation were shut down for 3 months, and subsequently limited to critical research operation in shifts for 9 months) have not allowed us to repeat every experiment multiple times. Instead, we included Figure 1—figure supplement 2B to demonstrate the reproducibility of the experiments.

The authors report evidence for disulfide bond formation between neighbored molecules at C394. The level of disulfide bond formation is known to depend on cofactors including protein concentration, oxygen exposure, pH, the presence of oxidizing or reducing agents, temperature, time, etc. Were those variables controlled?

Thank you for your questions. We have not done experiments to conclusively demonstrate the dependence of disulfide bond formation on protein concentration. On the other hand, we kept oxygen exposure, pH, temperature, and time consistent across the experiments. Regarding the presence of oxidizing or reducing agents, Figure 2B demonstrates that TCEP can be used to destabilize A_2A_R dimers.

Does the deca-His tag influence oligomerization of the protein?

Thank you for your question. We understand that since the study emphasizes the impact of the C-terminus, any modifications should be carefully considered, including the deca-His tag. However, we do not expect the His tag to majorly engage in inter-A_2A_R interactions in the absence of metal cations.

Does truncation of the receptor influence its function? Did the authors observe differences in ligand binding affinity and G protein activation rates for truncated/mutated receptor? Does protein truncation influence expression yield? Does truncation influence thermal stability of the expressed protein? Are eluted protein fractions obtained by size exclusion chromatography ligand binding- and G protein activation competent?

We understand that a study of function of truncated A_2A_R and its different oligomeric species is a priority. We are currently developing systems and tools that can serve as functional readouts for A_2A_R variants reconstituted in vitro.

Nevertheless, we have answers to some of the questions:

– We do not know yet whether truncation of the receptor influence A_2A_R function.

– Using densitometry on Western blots of XAC inactive and active fractions of different A_2A_R variants, we observed no significant difference in XAC affinity among the variants presented in this study. We do not have data on G protein activation rates.

– The expression yield is reduced upon truncation and mutation compared with the WT form but is still high enough for us to obtain enough for SEC analysis.

– We do not have data regarding the thermal stability of the truncated/mutated protein.

– We did not perform post-SEC functional analyses. However, since the protein was selected by a XAC ligand-affinity column prior to SEC, we believe that A_2A_R of the SEC eluent is capable of binding XAC.

Experimental results are accompanied by an impressive set of molecular simulations. Were simulations sufficiently long or repeated sufficiently often with variation of initial conditions to yield results that are not biased by initial conditions? Would it be possible to conduct a statistical analysis of properties of interaction sites between neighbored molecules? What is the role of protein segments other than the C-terminus for aggregation?

In response to the reviewer, we have carried out a decorrelation analysis to show that our simulations are not biased by our initial conditions. Our decorrelation time is about 4 µs, meaning that the systems are independent of the starting conformation after this point in time. With respect to interaction sites, our current analysis already captures all potential interaction sites, since the C-terminal segments are the portions of A_2A_R that predominantly form the dimerization interface. With respect to the heptahelical bundle of A_2A_R, based on our MD simulations, they play little or no role in receptor oligomerization since the length of the C-terminal tail sterically prohibits this interaction from taking place. Essentially, what this means is that the heptahelical bundles are not closer than 7 Å at any point in time in our simulations. This is a novel and distinct result from all previous MD simulations of A_2A_R (for example, Song et al., Biorxiv, https://doi.org/10.1101/2020.06.24.168260); because they did not include the full-length C-terminus of A_2A_R, dimerization interactions were observed between the TM helices of each protein.

**Author response image 1. respfig1:** Decorrelation time calculated from representative trajectories of each A_2A_R truncated system.

Reviewer #3 (Recommendations for the authors):1. The putative disulfide bridge involving C394 should be further investigated.–In light of the suggested C-terminus/C-terminus interaction in absence of the TM domain, the Cys partner might be in the C-terminus. It would be useful to provide a list of Cys residue that potentially can interact with Cys394 and experimentally validate these hypotheses.

We now include a list of cysteine residues that potentially can interact with C394 (namely residues C28, C82, C128, C185, C245, C254 in the TM domain). However, only C394 on the C-terminus is fully solventexposed, while the other cysteines are thought to be buried within the TM region of A2A. Experimental validation of these hypotheses involves repeating an exhaustive list of experiments for six different variants containing single mutants to test their potential interaction with C394. The pandemic has limited our time and resources such that such experiments could prove difficult (UCSB research operation was shut down for 3 months, and subsequently limited to critical research operation in shifts for 9 months).

An analysis of our MD simulations to monitor residue-residue distances between cysteines could be done as a secondary validation. Unfortunately, we do not expect them to be revealed in simulations since they only stabilize A_2A_R oligomers by kinetically trapping them after they are formed, as suggested in response to Reviewer #1 (Figure 5—figure supplement 1). A distance-based analysis for all potential cysteine pairs showed a complete lack of interactions that could potentially lead to formation of a disulfide bond (i.e., < 7 Å).

– The discussion in lines 338-391 should be extended accordingly: 'A previous study showed that residue C394 in A2AR dimer is available for nitroxide spin labelling(Schonenbach et al. 2016), suggesting that some of these disulfide bonds may be between 390 residue C394 and another cysteine in the hydrophobic core of A2AR that do not form intramolecular disulfide bonds(De Filippo et al. 2016; Naranjo et al. 2015; O'Malley et al. 2010)'

Thank you for your suggestion. We have modified the manuscript accordingly.

– Moreover, in some parts of the manuscript the putative disulfide bridge is ignored, es: Lines 86-87: 'a model GPCR that could engage in diverse non-covalent interactions, such as electrostatic interactions, hydrogen bonds, or hydrophobic interactions. These non-covalent interactions are readily tunable by external factors', Lines 291-293: 'The variable 13 291 nature of A2AR oligomeric interfaces suggests that the main driving forces must be non-covalent interactions, such as electrostatic interactions and hydrogen bonds as identified by the above MD simulations'.

Thank you for your comment. We indeed found that disulfide bonds are an important, non-negligible, force that can stabilize A_2A_R oligomers. However, we decided not to emphasize its role as a key driving force for A_2A_R dimerization/oligomerization for a few reasons. First, breaking off all disulfide bonds with TCEP only partially destabilized A_2A_R oligomers, while a significant dimer/oligomer population persisted (Figure 2B). Second, disulfide bonds should not be modulated by varying ionic strengths that, however, we have demonstrated directly alters the extent of A_2A_R oligomerization. Third, we isolated dimer/oligomer population of truncated A_2A_R and resuspended it in solution and subjected it to a second round of SEC, as described earlier. We found the dimer/oligomer population of A_2A_R lacking C394 to be equally stable and behave similarly as the A_2A_R WT population (Figure 5—figure supplement 1A). Given that, we can confidently conclude that disulfide bond formation is not the major driver of A_2A_R dimer/oligomer formation and stability.

It is also worth noting that the cytoplasm lacks the conditions and machinery required for disulfide bond formation, and hence disulfide formation is less important in the cellular context. Only few cases have been reported where cytoplasmic disulfide bonds are formed (Saaranen et al., Antioxidants and Redox Signaling 2013, 19 (1), 46–53; Locker et al., J Cell Biol 1999, 144 (2), 267–279), but how that occurs remains unknown.

2. MD simulationsThe C-terminus is not present in any of the A2AR crystal structures and is very long (Lines 104-105: A striking example is A2AR, a model GPCR with a particularly long, 122-residue, C-terminus that is truncated in all published structural biology studies).The C-terminus is therefore modelled, however, the only reference to the C-terminus modelling I could find is in Lines 594-595: 'missing residues added using MODELLER 9.23 (Eswar et al. 2006)'. Which template(s) was(were) used for the modelling and which is the sequence similarity? The detailed modelling procedure and the computational evaluation should be provided.

Thank you for your suggestion. To clarify, we only used the 5G53 structure as our template, and any additional residues beyond the C-terminus of the 5G53 sequence were generated by MODELLER. No other template structures were used. We have updated the methods section in the manuscript to reflect this. Since there is no biophysical data about the conformation of the C-terminus, we allowed MODELLER to generate a “best guess” conformation, followed by long timescale (microseconds) equilibrium MD simulations to allow the C-termini to sample all possible conformations. Please refer to our earlier response to reviewer #1 for additional details.

Results of the MD are highly dependent of the input model. Moreover, the information about the disulfide bridge is not incorporated in the models but this is an important structural feature to be considered.

We completely agree with the reviewer on the dependence of the input model. Explicitly modeling disulfide bridges in a coarse-grained system is inherently difficult and may introduce uncertainty in sampling of the dimer interface. Since we do not have any detailed structural information on the monomer-monomer interactions of A_2A_R that could inform potential orientations that facilitate disulfide bridge formation, we focused rather on modeling fully non-bonded monomer-monomer interactions. Other issues with the input model have either already been outlined in the methods or addressed in Rev. #1’s comments.

Also, the conclusions about the role of the ERR motif are based on the modelling, but we do not have information to judge the modelling.

We agree with the reviewer and have done our best to address this issue with respect to analysis of RMSD and RMSF. Please see our response(s) above.

Lines 409-410: 'This observation is supported by our experimental results showing that substituting this charged cluster with alanines reduces the total A2AR oligomer levels' – the experimental results suggest the involvement of these residues on the oligomerization process, but do not say a lot about the molecular mechanisms – localizing these residues far from the interacting surface and the intramolecular interactions are hypotheses based on the modelling.

Thank you for your suggestion. We agree that studying the molecular mechanisms of receptor oligomerization is critical in understanding and controlling this process. However, we think such an experimental study requires a lot of efforts and is out of the scope of this paper.

3. The impact of findings is weakly stated, some related sentences in the paper are very general:– Line 38 in the abstract: 'offering important guidance for structure-function studies of A2AR and other GPCRs'– Lines 55-56: 'it is crucial to identify the driving factors that govern the oligomerization of GPCRs, such that the properties of GPCR oligomers can be understood'– Lines 473-475: 'In that context, this study offers valuable insights and approaches to tune the oligomerization of A2AR and potentially of other GPCRs using its intrinsically disordered C-terminus'

Thank you for your suggestions. We have amended the manuscript as follows:

– Line 37–38 in the abstract: ‘offering important guidance on how to modify the C-terminus and tune receptor oligomerization for structure-function studies of A_2A_R and other GPCRs’.

– Line 55–56: ‘it is crucial to identify the driving factors of GPCR oligomerization, such that this process can be more deliberately controlled to facilitate structure-function studies of GPCRs.’

– Line 502–505: ‘In that context, this study offers valuable insights and approaches into how the oligomerization of A_2A_R and potentially of other GPCRs can be tuned by modifying the intrinsically disordered C-terminus and varying salt types and concentrations.’

4. I suggest labeling TM residues with BW numbering, so it will be easier to distinguish between TM residues and C-terminus residues in the figures and the text.

Thanks for the suggestion. We have modified the manuscript.

5. Title: 'homo-oligomerization' should replace 'oligomerization'.

Thank you. We have adjusted the title.